# A transcriptomics resource reveals a transcriptional transition during ordered sarcomere morphogenesis in flight muscle

Maria L Spletter[1,2]*, Christiane Barz[1], Assa Yeroslaviz[3], Xu Zhang[1,4,5], Sandra B Lemke[1], Adrien Bonnard[4,6], Erich Brunner[7], Giovanni Cardone[8], Konrad Basler[7], Bianca H Habermann[3,4,6], Frank Schnorrer[1,4]*

[1]Muscle Dynamics Group, Max Planck Institute of Biochemistry, Martinsried, Germany; [2]Biomedical Center, Physiological Chemistry, Ludwig-Maximilians-Universität München, Martinsried, Germany; [3]Computational Biology Group, Max Planck Institute of Biochemistry, Martinsried, Germany; [4]Aix Marseille Univ, CNRS, IBDM, Marseille, France; [5]School of Life Science and Engineering, Foshan University, Guangdong, China; [6]Aix Marseille Univ, INSERM, TAGC, Marseille, France; [7]Institute of Molecular Life Sciences, University of Zurich, Zurich, Switzerland; [8]Imaging Facility, Max Planck Institute of Biochemistry, Martinsried, Germany

**\*For correspondence:**
maria.spletter@bmc.med.lmu.de (MLS);
frank.schnorrer@univ-amu.fr (FS)

**Competing interests:** The authors declare that no competing interests exist.

**Abstract** Muscles organise pseudo-crystalline arrays of actin, myosin and titin filaments to build force-producing sarcomeres. To study sarcomerogenesis, we have generated a transcriptomics resource of developing *Drosophila* flight muscles and identified 40 distinct expression profile clusters. Strikingly, most sarcomeric components group in two clusters, which are strongly induced after all myofibrils have been assembled, indicating a transcriptional transition during myofibrillogenesis. Following myofibril assembly, many short sarcomeres are added to each myofibril. Subsequently, all sarcomeres mature, reaching 1.5 µm diameter and 3.2 µm length and acquiring stretch-sensitivity. The efficient induction of the transcriptional transition during myofibrillogenesis, including the transcriptional boost of sarcomeric components, requires in part the transcriptional regulator Spalt major. As a consequence of Spalt knock-down, sarcomere maturation is defective and fibers fail to gain stretch-sensitivity. Together, this defines an ordered sarcomere morphogenesis process under precise transcriptional control – a concept that may also apply to vertebrate muscle or heart development.
DOI: https://doi.org/10.7554/eLife.34058.001

## Introduction

Sarcomeres are the stereotyped force producing mini-machines present in all striated muscles of bilaterians. They are built of three filament types arrayed in a pseudo-crystalline order: actin filaments are cross-linked with their plus ends at the sarcomeric Z-disc and face with their minus ends towards the sarcomere center. In the center, symmetric bipolar muscle myosin filaments, anchored at the M-line, can interact with the actin filaments. Myosin movement towards the actin plus ends thus produces force during sarcomere shortening. Both filament types are permanently linked by a third filament type, the connecting filaments, formed of titin molecules (*Gautel and Djinović-Carugo, 2016*; *Lange et al., 2006*). A remarkable feature of sarcomeres is their stereotyped size, ranging from 3.0 to 3.4 µm in relaxed human skeletal muscle fibers (*Ehler and Gautel, 2008*; *Llewellyn et al., 2008*; *Regev et al., 2011*). Even more remarkable, the length of each bipolar myosin filament is 1.6 µm in all mature sarcomeres of vertebrate muscles, requiring about 300 myosin hexamers to assemble per filament (*Gokhin and Fowler, 2013*; *Tskhovrebova and Trinick, 2003*).

**eLife digest** Animals may have different types of muscles but they all have one thing in common: molecular machines called sarcomeres that produce a pulling force. Conserved from fruit flies to humans, these structures line up end-to-end inside muscle cells, forming long cables called myofibrils. Some of the myofibrils in a human can reach several centimetres in length, which is much longer than those in a fruit fly. However, individual sarcomeres are the same length in both humans and flies.

To build the parts of the sarcomere, an animal cell first copies the relevant genes into intermediate molecules known as mRNAs, which are then translated to build new sarcomere proteins. Developing muscle cells later tune their sarcomeres to make them sensitive to stretching. This tweaks the power and force of the mature muscle, but the details of this developmental process are not fully understood.

Now, Spletter et al. have counted all the mRNAs in the developing flight muscles of fruit flies, with the aim of generating a resource that catalogues the changes in gene activity, or expression, that occur as muscles develop. This revealed that sarcomeres form in three phases. First, the cells assembled all their myofibrils. Then, they added short sarcomeres to the ends of their myofibrils. Finally, the sarcomeres matured to their full length and diameter, and became sensitive to stretching.

Fruit fly muscles had 40 patterns of gene expression, with most of the sarcomere components having one of two specific patterns. The expression of these genes dramatically rose after the young muscle cells had finished assembling all their myofibrils, suggesting muscles express different genes when their sarcomeres mature. A protein called spalt-major helped the cell to know when to make the transition, allowing the sarcomeres to grow in length and width.

Losing spalt-major late in muscle development stopped sarcomere growth and prevented the tuning process. The sarcomeres failed to become sensitive to stretching, a crucial feature of mature muscle. Muscles without spalt-major contracted too much and without coordination, like a muscle spasm.

The similarities between fruit fly and human sarcomeres suggest this developmental sequence may also occur in human muscles too. Understanding these steps may help to improve repair after injury or muscle growth during exercise. The next step is to test whether regenerating or growing muscles develop in the same way.

DOI: https://doi.org/10.7554/eLife.34058.002

Human muscle fibers can be several centimetres in length and both ends of each fiber need to be stably connected to tendons to achieve body movements. As sarcomeres are only a few micrometres in length, many hundreds need to assemble into long linear myofibrils that span from one muscle end to the other and thus enable force transmission from the sarcomeric series to the skeleton (*Lemke and Schnorrer, 2017*). Thus far, we have a very limited understanding of how sarcomeres initially assemble into long immature myofibrils during muscle development to exactly match the length of the mature muscle fiber (*Sparrow and Schöck, 2009*). In particular, we would like to understand how such sarcomeres mature to the very precise stereotyped machines present in mature muscle fibers.

Across evolution, both the pseudo-crystalline regularity of sarcomeres as well as their molecular components are well conserved (*Ehler and Gautel, 2008*; *Vigoreaux, 2006*). Thus, *Drosophila* is a valid model to investigate the biogenesis of sarcomeres as well as their maturation. In particular, the large indirect flight muscles (IFMs) that span the entire fly thorax are an ideal model system to investigate mechanisms of myofibrillogenesis. They contain thousands of myofibrils consisting of 3.2 μm long sarcomeres (*Schönbauer et al., 2011*; *Spletter et al., 2015*).

Like all *Drosophila* adult muscles, IFMs are formed during pupal development from a pool of undifferentiated myoblasts called adult muscle precursors (AMPs) (*Bate et al., 1991*). From 8 hr after puparium formation (APF), these AMPs either fuse with themselves (for the dorso-ventral flight muscles, DVMs) or with remodelled larval template muscles (for the dorso-longitudinal flight muscles, DLMs) to form myotubes (*Dutta et al., 2004*; *Fernandes et al., 1991*). These myotubes

develop dynamic leading edges at both ends and initiate attachment to their respective tendon cells at 12 to 16 hr APF (*Weitkunat et al., 2014*). These attachments mature and mechanical tension is built up in the myotubes, followed by the formation of the first immature periodic myofibrils at 30 hr APF when the muscle fibers are about 150 μm in length. These immature myofibrils contain the earliest sarcomeres, which are about 1.8 μm in length (*Weitkunat et al., 2014*). During the remaining 3 days of pupal development, the muscle fibers grow to about 1 mm to fill the entire thorax and sarcomere length increases to a final length of about 3.2 μm in adult flies (*Orfanos et al., 2015*; *Reedy and Beall, 1993*).

After myoblasts have fused to myotubes, the flight muscle specific selector gene *spalt major* (*spalt, salm*) is turned on in the developing flight muscle myotubes. Spalt major is responsible for the correct fate determination and development of the flight muscles, which includes the fibrillar flight muscle morphology and the stretch-activated muscle contraction mode (*Schönbauer et al., 2011*; *Syme and Josephson, 2002*). It does so by controlling the expression of more than 700 flight muscle specific genes or gene isoforms during development (*Spletter and Schnorrer, 2014*; *Spletter et al., 2015*). However, how the interplay between all these isoforms instructs the formation of highly regular, pseudo-crystalline sarcomeres in the flight muscle is not understood.

Here, we studied the transcriptional dynamics of flight muscle development in detail. We performed a systematic mRNA-Seq time-course of isolated muscle tissue at eight time points from the myoblast stage until the mature adult muscle stage, generating a transcriptomics resource of developing flight muscle. Bioinformatic analysis of expression dynamics identified two gene clusters that are strongly enriched for sarcomeric genes. The temporal dynamics of these clusters identified a transcriptional transition that is required for sarcomere morphogenesis. We define sarcomere morphogenesis in three sequential although overlapping phases. First, immature myofibrils assemble simultaneously; second, short sarcomeres are added to each myofibril; and third, all sarcomeres mature to their final length and diameter and acquire stretch-sensitivity. Interestingly, the number of myofibrils remains constant, suggesting that every sarcomere progresses through sarcomere maturation. We show that the flight muscle selector gene *spalt major* contributes to the observed transcriptional transition, suggesting that muscle fiber type-specific transcription is continuously required during sarcomere formation and maturation. Together, these findings indicate that precise transcriptional control of the sarcomeric components enables their ordered assembly into sarcomeres and their maturation to pseudo-crystalline regularity.

## Results

### A time-course of indirect flight muscle development

To better understand muscle morphogenesis in general and myofibrillogenesis in particular, we focused on the *Drosophila* indirect flight muscles (IFMs). We hypothesised that major morphological transitions during IFM development may be induced by transcriptional changes, thus we aimed to generate a detailed developmental mRNA-Seq dataset from IFMs. IFMs are built from AMPs that adhere to the hinge region of the wing disc epithelium and are labelled with GFP-Gma under *Him* control (*Figure 1A*) (*Soler and Taylor, 2009*). At 16 hr APF, many of these myoblasts have fused to larval template muscles to build the dorsal-longitudinal flight muscle (DLM) myotubes, which initiate attachment to their tendons. At this stage, the DLM myotubes of fibers 3 and 4 have a length of about 300 μm (*Figure 1B*). Fusion ceases at about 24 hr APF (*Figure 1C*) and attachment matures until 32 hr APF, coinciding with the strong recruitment of βPS-Integrin and the spectraplakin homolog Shortstop (Shot) to the attachment sites. At this stage the myofibers have built up mechanical tension and compacted to a length of about 150 μm, coinciding with the appearance of long Shot-positive tendon extensions that anchor the muscles within the thorax. This important developmental transition is highlighted by the assembly of immature myofibrils visualised by strong F-actin staining throughout the entire muscle fiber (*Figure 1D*) (*Weitkunat et al., 2014*).

After 32 hr APF, the myofibers undergo another developmental transition and begin to grow dramatically. They elongate 3-fold to reach a length of about 480 μm by 48 hr APF (*Figure 1E*) and about 590 μm by 56 hr APF (*Figure 1F*). Concomitantly, the tendon extensions shrink with the myofibers being directly connected to the basal side of the tendon cell epithelium by 72 hr APF

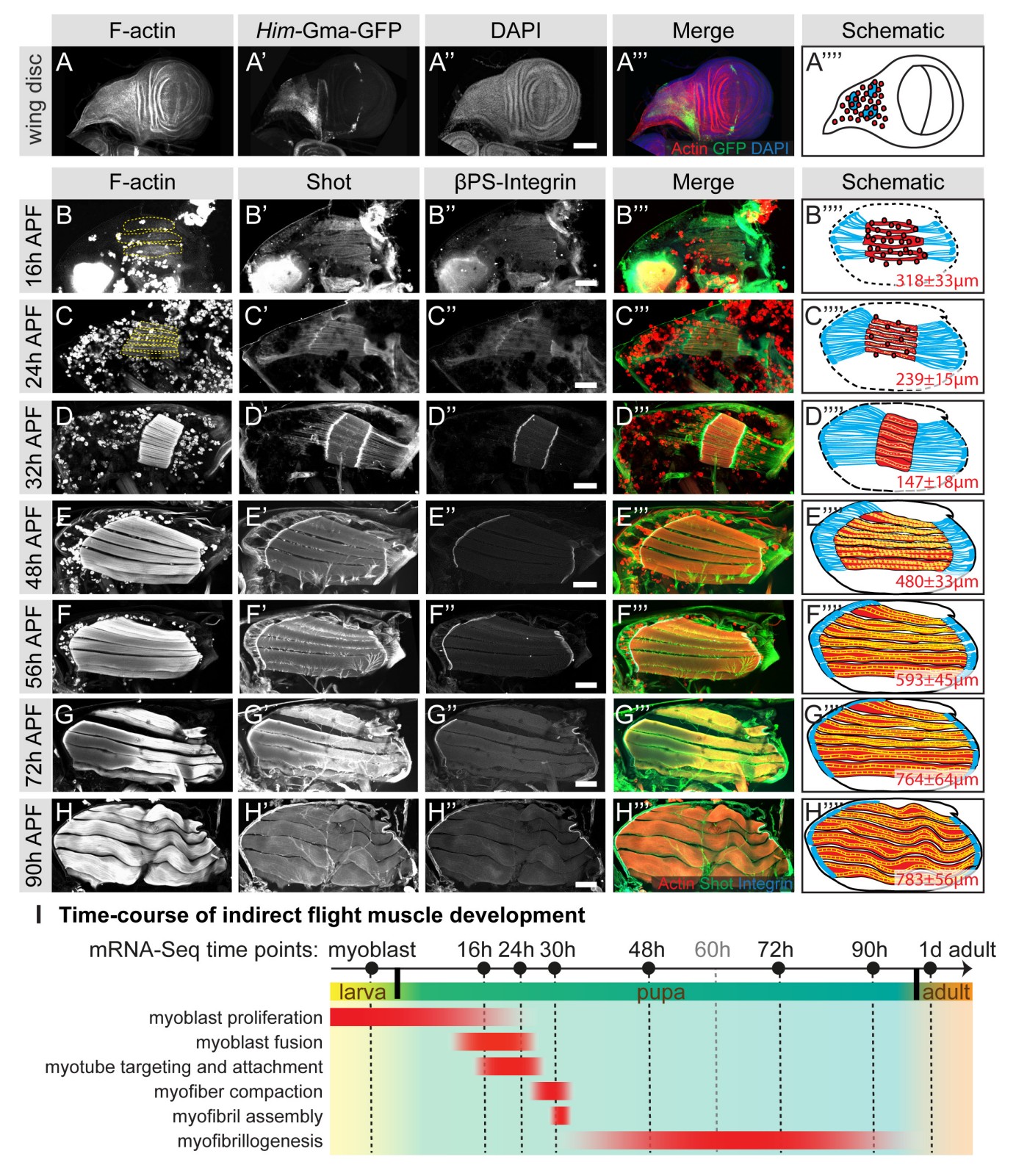

**Figure 1.** Development of the dorsal longitudinal indirect flight muscles. (**A–H**) Time-course of DLM development. (**A**) Myoblasts adhering to the hinge of the larval wing disc were visualised with *Him*-GFP-Gma (green), F-actin was stained with phalloidin (red) and nuclei with DAPI (blue). (**B–H**) Time-course of DLM and myotendinous junction development at 16 hr (**B**), 24 hr (**C**), 32 hr (**D**), 48 hr (**E**), 56 hr (**F**), 72 hr (**G**) and 90 hr APF (**H**). F-actin was stained with phalloidin (red), Shot (green) and βPS-Integrin (blue). DLMs are highlighted in dashed yellow lines in B an C. (**A'''–C'''**) Scheme of the

*Figure 1 continued on next page*

Figure 1 continued

respective developmental stages with myoblasts and muscles in red, tendon cells in blue and wing disc or pupal thorax outline in black. The length of the muscle fibers in indicated in red. For details see text. Scale bar represents 100 μm. (I) Temporal summary of known events during myogenesis (red). Samples for mRNA-Seq were collected at time points noted in black.

DOI: https://doi.org/10.7554/eLife.34058.003

The following source data is available for figure 1:

**Source data 1.** This table includes the length measurements of the indirect flight muscle fibers at the indicated stage.

DOI: https://doi.org/10.7554/eLife.34058.004

(*Figure 1G*). At the end of pupal development (90 hr APF), wavy muscle fibers with a length of about 780 μm containing mature myofibrils (*Figure 1H*) are present within the thorax.

## A transcriptomics resource of indirect flight muscle development

To quantify transcriptional dynamics across the entire developmental time-course, we focused on the major developmental transitions and isolated mRNA from dissociated myoblasts of dissected or mass-isolated third instar wing discs and from hand-dissected IFMs at 16, 24, 30, 48, 72 and 90 hr APF pupae, and adult flies 1 day after eclosion (*Figure 1I*). We performed mRNA-Seq using at least two biological replicates for each time point (see Materials and methods). To identify genes with similar temporal expression profiles, we used Mfuzz (*Kumar and Futschik, 2007*) to cluster standard normalized read counts from all genes expressed above background (12,495 of 13,322 genes). This allowed us to confidently identify 40 distinct genome-wide clusters (*Figure 2—figure supplement 1*), each of which contains a unique gene set ranging from 155 to 703 members (*Supplementary file 1*). These clusters represent various temporal expression dynamics, with high expression at early (myoblast proliferation and fusion), mid (myotube attachment and myofibril assembly) or late (myo-fiber growth) myogenesis stages or a combination thereof (*Figure 1I*, *Figure 2—figure supplement 1*). These distinct patterns suggest a precise temporal transcriptional regulation corresponding to observed morphological transition points.

To verify the mRNA-Seq and cluster analysis, we selected a number of 'indicator' genes with available antibodies or GFP fusion proteins whose expression correlates with important developmental transitions. Twist (Twi) is a myoblast nuclear marker at larval stages and its expression needs to be down-regulated after myoblast fusion in pupae (*Anant et al., 1998*). We find *twi* mRNA in Mfuzz cluster 27, with high expression in myoblasts until 16 hr APF and a significant down-regulation from 24 hr APF, which we were able to verify with antibody stainings (*Figure 2A–C*). The flight muscle fate selector gene *spalt major* (*salm*) (*Schönbauer et al., 2011*) and its target, the IFM splicing regulator *arrest* (*aret, bruno*) (*Spletter et al., 2015*) are members of cluster 26 and 14, respectively. Expression of both clusters is up-regulated after myoblast fusion at 16 hr APF, which we were able to verify with antibody stainings (*Figure 2D–F*, *Figure 2—figure supplement 2A–C*). The apparent *salm* mRNA peak at 72 hr APF, which does not appear to cause a further protein increase, represents a mere 1.3 fold increase in expression and would need further confirmation as Salm, like many transcription factors, is expressed at low levels. For the initiation of muscle attachment, we selected Kon-tiki (Kon) (*Schnorrer et al., 2007*; *Weitkunat et al., 2014*), member of cluster 15, which is transiently up-regulated after myoblast fusion, before it is down-regulated again after 30 hr APF. Consistently, we found Kon-GFP present at muscle attachment sites at 30 hr APF but not at 72 hr APF (*Figure 2G–I*). A similar expression peak shifted to slightly later time points is found in cluster 34, which contains β-*tubulin 60D* (*βTub60D*) (*Leiss et al., 1988*). Consistently, we find β-Tub60D-GFP (*Sarov et al., 2016*) expression in IFMs at 30 and 48 hr but not 72 hr APF (*Figure 2J–L*). After attachment is initiated, the attachments need to mature and be maintained. As expected, we found the essential attachment components βPS-Integrin (*mys*) and Talin (*rhea*) in clusters that are up-regulated after myoblast fusion until adulthood (clusters 7 and 25, respectively). This is consistent with continuous high protein expression of βPS-Integrin-GFP and Talin-GFP at muscle attachment sites (*Figure 2M–O*, *Figure 2—figure supplement 2D–F*). Taken together, these semi-quantitative protein localisation data nicely validate the temporal mRNA dynamics found in the mRNA-Seq data, confirming our methodology.

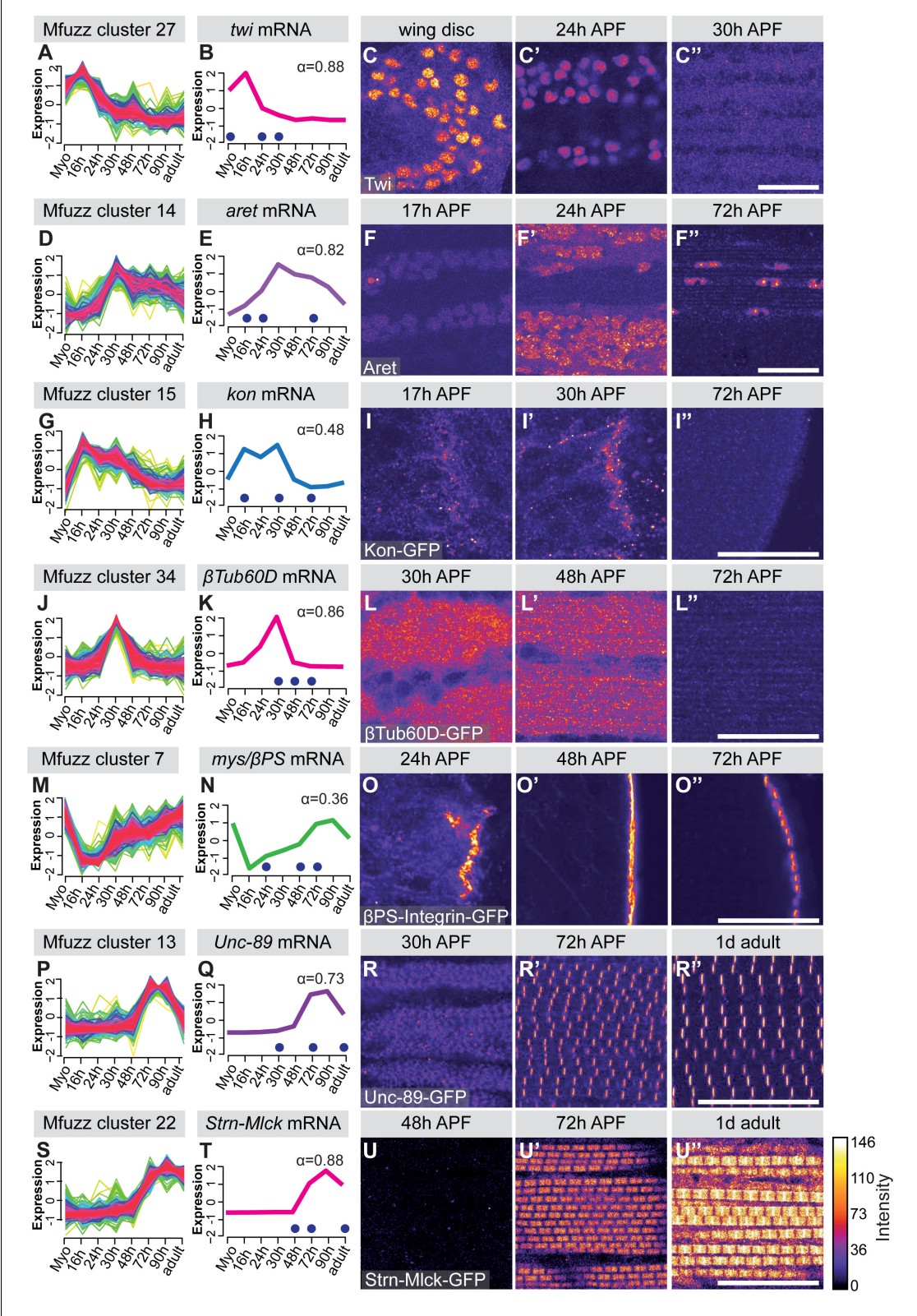

**Figure 2.** Verification of mRNA-Seq time-course by 'indicator' gene expression. (**A,D,G,J,M,P,S**) Temporal expression dynamics were evaluated by clustering standard-normal mRNA-seq counts using Mfuzz. Temporal expression profiles are plotted with high membership values in warm colours (red, pink), and lower membership values in cool colours (blue, green). (**B,E,H,K,N,Q,T**) The profile of one 'indicator' gene from each cluster is shown and coloured based on the Mfuzz membership value α. (**C,F,I,L,O,R,U**). Protein expression and localisation dynamics were visualised by antibody staining

*Figure 2 continued on next page*

*Figure 2 continued*

against Twi (C) and Aret (F) or against GFP for GFP tagged fosmid reporters for Kon (I), β-Tub60D (L), βPS-Integrin (*mys*) (O), Unc-89 (Obscurin) (R) and Strn-Mlck (U). Images for the same protein were acquired using the same settings, and pseudo-coloured based on intensity. Note the close correlation between mRNA and protein expression dynamics. Time points are indicated by blue dots on the mRNA expression profile. Scale bars represent 20 μm.

DOI: https://doi.org/10.7554/eLife.34058.005

The following figure supplements are available for figure 2:

**Figure supplement 1.** Forty distinct temporal mRNA-Seq expression profiles.

DOI: https://doi.org/10.7554/eLife.34058.006

**Figure supplement 2.** Additional examples of 'indicator' gene expression.

DOI: https://doi.org/10.7554/eLife.34058.007

## A transcriptional transition after 30 hr APF

Hierarchical clustering of the core expression profiles from the 40 identified Mfuzz clusters defines eight temporally ordered groups (*Figure 3*) that show progressive expression dynamics as muscle development proceeds. A time-dependent shift in gene ontology (GO) term enrichments is apparent between the eight groups, reflecting the different stages of IFM development. GO-Elite analysis (*Zambon et al., 2012*) for gene set enrichment identified GO terms related to cell proliferation and development as enriched in the early clusters (such as the *twi* cluster 27 or the *kon* cluster 15), whereas terms related to actin filament dynamics are more enriched in the middle clusters (such as the βPS-Integrin cluster seven and the Talin cluster 25), reassuring that the clustering approach is valid (*Figure 3*, *Supplementary file 2*). Strikingly, the only two clusters that display a strong enrichment for genes important for sarcomere organisation are clusters 13 and 22, both of which are late up-regulated clusters (*Figure 3*). Members of both clusters just become detectable at 30 hr APF (Unc-89/Obscurin-GFP, Act88F-GFP, Mhc-GFP) or even later at 48 hr APF (Strn-Mlck-GFP, Mf-GFP). In all cases, we could confirm the strong up-regulation from 30 hr to 72 hr APF in the mRNA-Seq data at the protein level using GFP fusion proteins under endogenous control (*Figure 2P–U*, *Figure 2—figure supplement 2J–P*) (*Sarov et al., 2016*). We additionally verified the late up-regulation of Flightin (Fln), a member of cluster 3, which is detectable at 72 hr but not at 30 hr APF (*Figure 2—figure supplement 2G–I*).

At late stages of flight muscle development, mitochondrial density strongly increases (*Clark et al., 2006*). Using GO-Elite, we found a strong enrichment for mitochondrial related pathways in four late up-regulated clusters, namely 3, 28, 39 as well as the sarcomere cluster 22 (*Figure 3*). By comparing the clusters to systematic functional data acquired at all stages of *Drosophila* muscle development (*Schnorrer et al., 2010*), we find enrichments in clusters throughout the time-course. Interestingly, genes highly expressed in flight muscle compared to other muscle types, identified as '*salm*-core genes' (*Spletter et al., 2015*), are also enriched in the late clusters, including the mitochondrial enriched clusters 3, 28, 39 and the sarcomere enriched cluster 22 (*Figure 3*). These data highlight the changes in biological process enrichments that parallel expression dynamics, with a particular transition happening during later stages of muscle development after 30 hr APF. This corresponds to a time period after immature myofibrils have been assembled, which thus far remained largely unexplored.

To examine the temporal expression dynamics in more detail, we performed a principle component analysis (PCA) of the mRNA-Seq time points and Mfuzz clusters and found that the major variance is developmental time, with a notable change after 30 hr APF (*Figure 4A*, *Figure 4—figure supplement 1A*). There are a large number of genes being up-regulated as well as down-regulated between 30 and 48 hr and between 48 and 72 hr APF, with major differences between the sets of genes expressed at early (16–30 hr APF) versus late (72–90 hr APF) stages of development (*Figure 4B*, *Figure 4—figure supplement 1B*). Thus, we focused our attention on the transcriptional transition between 30 and 72 hr APF, which correlates with major growth of the flight muscle fibers (*Figure 1*).

A large number of genes are significantly up- or down-regulated from 30 hr to 72 hr APF, as visualized on a volcano plot displaying $\log_2$fold changes (FC) (*Figure 4C*), suggesting a major change in gene expression. In particular, many sarcomeric genes are strongly up-regulated. To identify fine details in expression dynamics, we took all genes significantly up-regulated between 30 and 72 hr

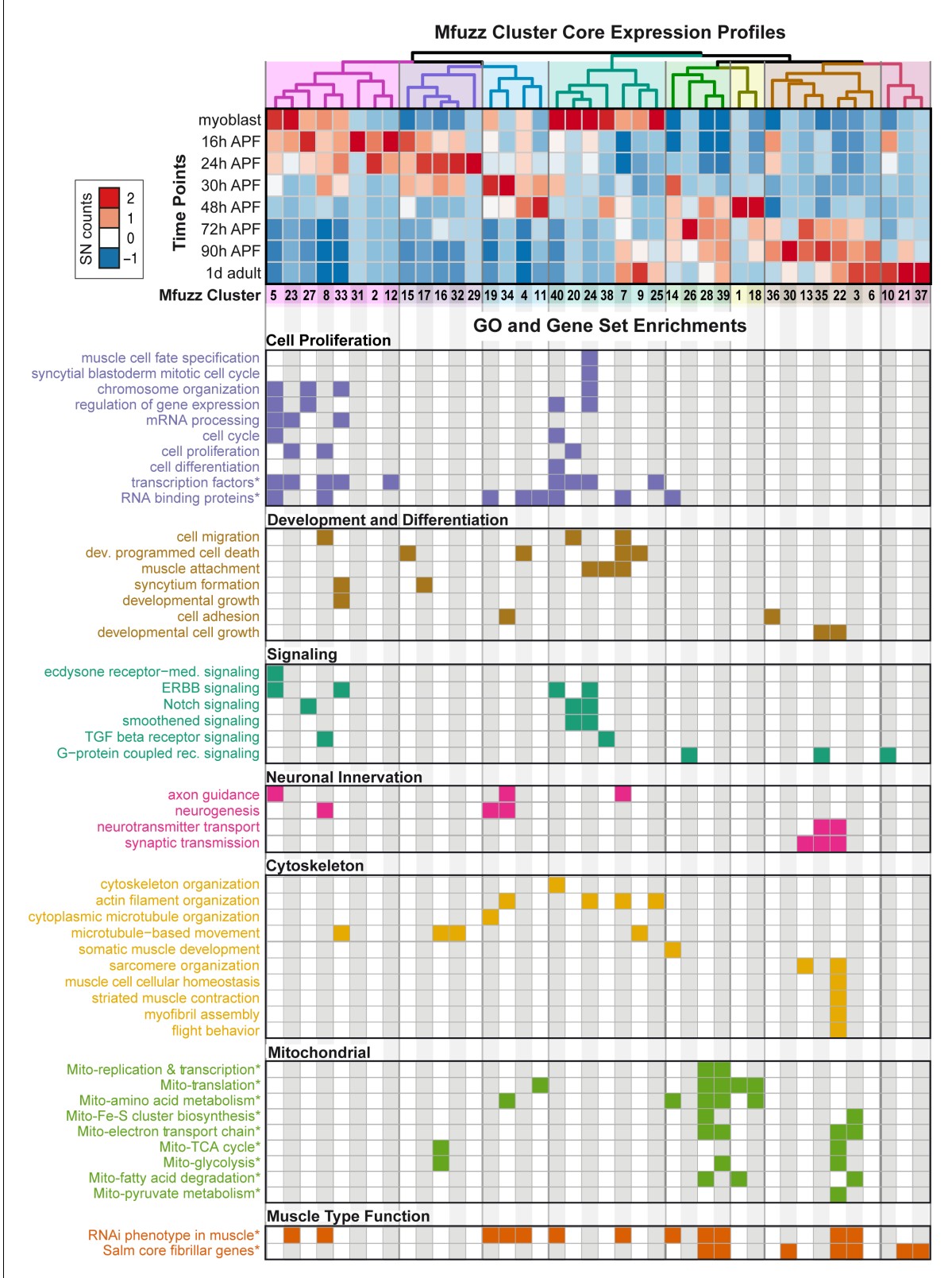

**Figure 3.** Expression dynamics reveal a temporal ordering of biological processes during muscle morphogenesis. (**Top**) Heat map of Mfuzz cluster core expression profiles. Standard-normal count values for all genes with a membership value α >0.8 were averaged to generate the core expression profile for each cluster. Mfuzz expression profiles fall into eight groups (coloured dendrogram leaves) based on hierarchical clustering of their temporal expression dynamics. Time points and Mfuzz clusters are labelled. Colour scale of standard-normal count values ranges from blue (stable/no

*Figure 3 continued on next page*

Figure 3 continued

expression) to red (high expression). (Bottom) GO Biological Process and user-defined gene set (marked with *) enrichments calculated with GO-Elite. Note that proliferation, development and differentiation terms are enriched at early time points, while mitochondrial and sarcomere terms are enriched at late time points. A coloured box indicates a significant enrichment of a given term in the specified cluster (see *Supplementary file 2* for details).

DOI: https://doi.org/10.7554/eLife.34058.008

APF and performed hierarchical clustering of their DESeq2 normalized counts values (*Figure 4D*). We noted that many genes are turned on from 30 hr to 72 hr APF, whereas others are already expressed at 30 hr and strongly increase their expression until 72 hr (*Figure 4D*), suggesting a transcriptional transition after 30 hr APF. Consistently, we found GO-terms of cell proliferation, cell cycle and Notch signalling down-regulated, whereas actin cytoskeleton, sarcomere, muscle function and mitochondrial related gene sets are strongly up-regulated from 30 hr to 72 hr APF (*Figure 4E*). Finally, the genes up-regulated from 30 hr to 72 hr APF are enriched for sarcomeric proteins and the '*salm* core genes' (*Figure 4—figure supplement 1C,D*), and the Mfuzz gene clusters with the most up-regulated members are the mitochondrial and both sarcomeric gene containing clusters 13 and 22 (*Figure 4F*). Together, these data suggest that in particular expression of the sarcomeric and mitochondrial genes is strongly induced after 30 hr APF.

## Ordered phases during sarcomere morphogenesis

The strong up-regulation of sarcomeric gene expression after immature myofibrils have been assembled (*Figure 1*) (*Weitkunat et al., 2014*) caught our interest and prompted us to more closely investigate the later stages of myofibrillogenesis during which myofibers grow dramatically (*Figure 1*). We stained the myofibers with phalloidin to reveal myofibril morphology and with the titin isoform Kettin (an isoform of the *sallimus* gene) to label the developing Z-discs and systematically quantified sarcomere length and myofibril width (see Materials and methods) (*Figure 5A*, *Supplementary file 3*). By measuring the total muscle fiber length, we calculated the total number of sarcomeres per myofibril at a given stage. We found that the sarcomere length and width remain relatively constant at about 2.0 and 0.5 µm, respectively until 48 hr APF (*Figure 5B,C*, *Supplementary file 3*). However, the number of sarcomeres per myofibril dramatically increases from about 100 at 34 hr to about 230 at 48 hr APF. After 48 hr only a few more sarcomeres are added, resulting in about 270 sarcomeres per myofibril at 60 hr APF. This number remains constant until the fly ecloses (*Figure 5D*, *Supplementary file 3*). Moreover, by analysing fiber cross-sections we found that the growth of the individual myofibril diameter correlates with growth of the entire muscle fiber. Both fiber diameter and myofibril diameter remain constant from 30 hr to 48 hr APF. After 48 hr the myofibril diameter grows nearly 3-fold from 0.46 µm to 1.43 µm in adult flies (*Figure 5E–G*), while fiber cross-sectional area grows nearly 4-fold from 1,759 µm$^2$ to 6,970 µm$^2$. Strikingly, during the entire time period from 30 hr APF to adults, the total number of myofibrils per muscle fiber remains constant (about 2000 per muscle fiber, *Figure 5H*). Taken together, these quantitative data lead us to propose ordered but somewhat overlapping phases of sarcomere morphogenesis: (1) During the sarcomere assembly phase, about 100 immature sarcomeres self-assemble within each immature myofibril. (2) Many short sarcomeres are added to each myofibril increasing its length. (3) The short sarcomeres grow in length and thickness to reach the mature pseudo-crystalline pattern. No new myofibrils are built after the initial myofibril assembly phase.

We gained additional evidence to support this ordered myofibrillogenesis model on both the molecular and functional levels. First, the initial assembly versus the later sarcomere maturation complement the transition in gene expression we observe from 30 hr to 72 hr APF. Using members of the late induced Mfuzz clusters that contain sarcomeric components, we found that indeed a subset of sarcomeric proteins, such as Unc-89/Obscurin and Mhc, are already detectable in a periodic pattern on immature myofibrils at 30 hr. By contrast, other important components, including Fln, Myofilin (Mf) and Strn-Mlck, are only incorporated into myofibrils from 48 hr APF, showing high levels by 72 hr APF (*Figure 5—figure supplement 1*). Second, to investigate muscle function, we used Talin-YPet as a muscle attachment marker and quantified the number of spontaneous muscle contractions in intact pupae (see Materials and methods). Interestingly, we found that immature myofibrils already start to spontaneously contract at 30 hr APF. These spontaneous contractions increase in strength

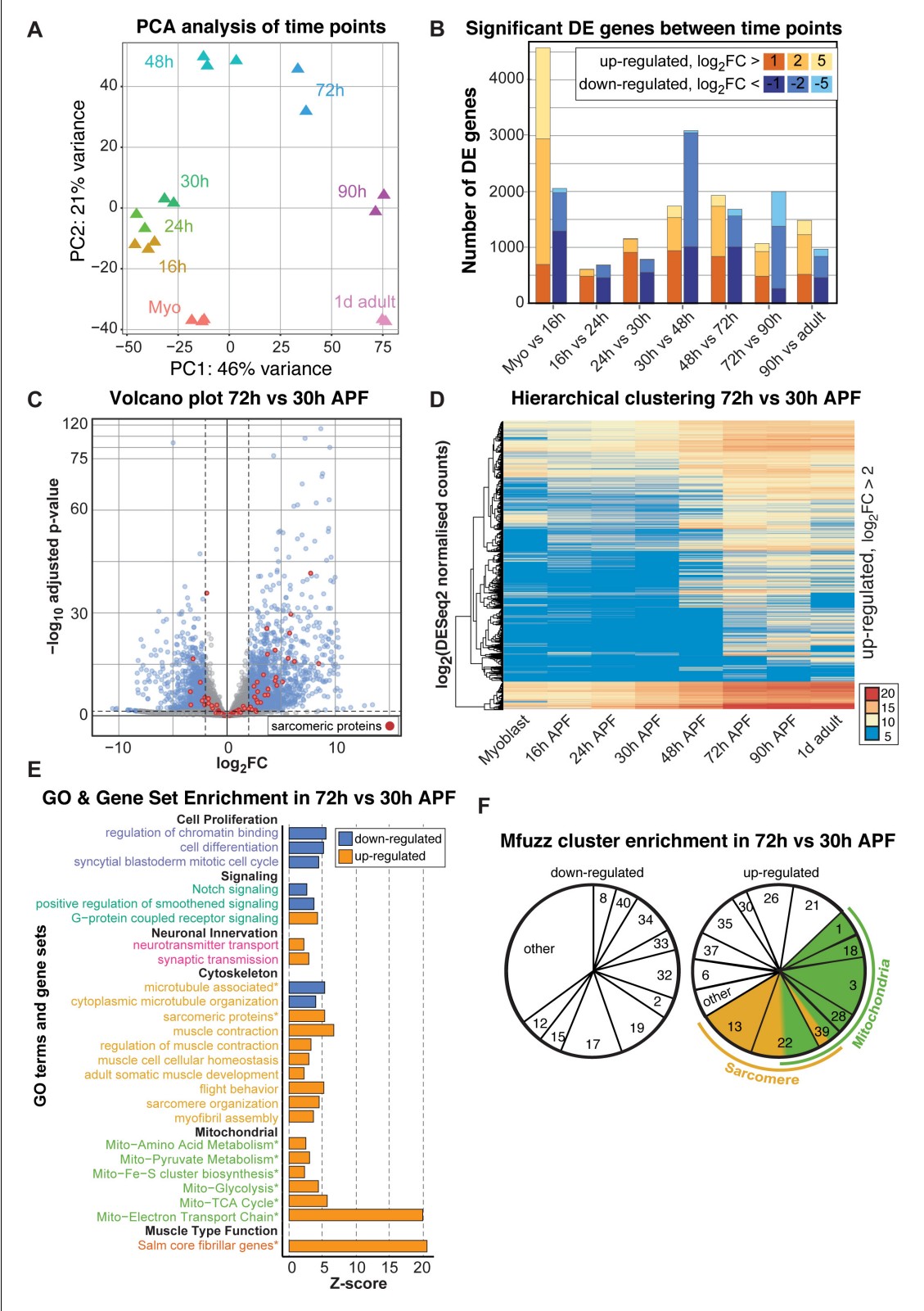

**Figure 4.** A major transition in gene expression after 30 hr APF. (**A**) DESeq2 principle components analysis (PCA) of all mRNA-Seq libraries. Each triangle represents a different biological replicate coloured by time point. Note that individual replicates for a given time point cluster together. PC1 divides early (≤48 hr APF) from late (≥72 hr) stages. (**B**) Stacked box plot of the number of significantly differentially expressed genes up-regulated (reds) or down-regulated (blues) between sequential time points with a p-value<0.05 and a log$_2$FC (fold change) of >1 (dark), >2 (medium) or >5 (light).

*Figure 4 continued on next page*

*Figure 4 continued*

The large differences between myoblast to 16 hr APF reflect muscle specification. A second large shift in expression is evident between 30 and 72 hr APF. (**C**) Volcano plot illustrating the strong up-regulation of sarcomeric proteins (red) from 30 hr to 72 hr APF. Significantly up- or down-regulated genes are in blue (p-value<0.05 and abs(log$_2$FC)>2). (**D**) Hierarchical clustering of log$_2$ transformed DESeq2 normalized counts for all genes that are significantly up-regulated between 30 and 72 hr APF. Note that they are either strongly induced at 48 or 72 hr APF (from yellow to red), or only turned on at 48 or 72 hr APF (blue to yellow/red), suggesting a major transition in gene expression after 30 hr APF. Colour scale of log$_2$ count values ranges from blue (not expressed) to red (highly expressed). (**E**) GO-Elite and user-defined gene set (marked with *) enrichments in up- (red) and down- (blue) regulated genes from 30 hr to 72 hr APF. Note the strong enrichment of mitochondrial and sarcomere terms in the up- regulated genes. (**F**) Pie charts showing the proportion of genes belonging to an enriched Mfuzz cluster in the sets of genes either up- or down-regulated from 30 hr to 72 hr APF. Note that a large proportion of genes up-regulated 30 hr to 72 hr belong to cluster 22, as well as Mfuzz clusters enriched for sarcomere (yellow) or mitochondrial (green) terms.

DOI: https://doi.org/10.7554/eLife.34058.009

The following figure supplement is available for figure 4:

**Figure supplement 1.** Additional evidence supporting a transition in gene expression between 30 and 72 hr APF.

DOI: https://doi.org/10.7554/eLife.34058.010

and frequency until 48 hr APF, but then cease, producing no detectable spontaneous contractions at 72 hr APF (*Figure 5I,J*, *Figure 5—video 1*). This demonstrates that during the sarcomere assembly phase, immature contractile myofibrils are generated, which then likely acquire stretch-sensitivity as the immature myofibrils mature and thus cease contracting.

## Salm contributes to the transcriptional transition after 30 hr APF

How does the transcriptional transition of the various sarcomeric components instruct myofibrillogenesis? As the identified sarcomeric clusters 13 and 22 are enriched for 'salm-core genes' (*Spletter et al., 2015*) (*Figure 3*), we chose to investigate gene expression in developing *spalt-major* knock-down (*salmIR*) flight muscles compared to wild type (*Supplementary file 4*). *salmIR* IFM shows a strong down-regulation in gene expression, notably of mRNAs coding for sarcomeric and mitochondrial components at 24 and 30 hr APF, and in particular at 72 hr APF (*Figure 6A–C*, *Figure 6—figure supplement 1A–D*). The genes down-regulated in *salmIR* IFM are enriched for GO terms associated with sarcomere assembly, flight behaviour and mitochondrial genes, as well as for the mitochondrial Mfuzz clusters 3, 28, 39 and the sarcomeric Mfuzz clusters 13 and 22 (*Figure 6—figure supplement 1E–G*). Interestingly, members of cluster 22, which is strongly enriched for sarcomeric and mitochondrial genes, are not only down-regulated at 72 hr APF in *salmIR* (*Figure 6B*) but are also less strongly induced from 30 hr to 72 hr APF in *salmIR* muscle compared to wild type (*Figure 6D*), suggesting that *salm* in addition to other factors is indeed required for the strong induction of sarcomeric protein expression after 30 hr APF.

Salm is expressed shortly after myoblast fusion and constitutive knock-down of *salm* with *Mef2*-GAL4 results in a major shift of muscle fiber fate (*Schönbauer et al., 2011*), which may indirectly influence transcription after 30 hr APF. Hence, we aimed to reduce Salm levels only later in development, to directly address its role during sarcomere maturation. To this end, we knocked-down *salm* with the flight muscle specific driver *Act88F*-GAL4, which is expressed from about 18 hr APF and requires *salm* activity for its expression (*Bryantsev et al., 2012*; *Spletter et al., 2015*). This strategy enabled us to reduce Salm protein levels at 24 hr APF resulting in undetectable Salm levels at 72 hr APF (*Figure 6—figure supplement 2*). To test if Salm contributes to the transcriptional boost of sarcomeric components after 30 hr APF, we performed quantitative imaging using unfixed living flight muscles expressing GFP fusion proteins under endogenous control. We used green fluorescent beads to normalise the GFP intensity between different samples, and could verify the induction of Mhc, Unc-89, Fln and Strn-Mlck on the protein level from 30 hr to 72 hr APF (*Figure 6—figure supplement 3*). While overall sarcomere morphology is not strongly affected in *Act88F >> salmIR* muscles, we found that the levels of Strn-Mlck, Fln and Mhc proteins are strongly reduced at 90 hr as compared to wild-type controls (*Figure 6D–H*). This suggests that *salm* indeed contributes to the transcriptional transition that boosts the expression of a number of sarcomeric proteins after 30 hr APF.

To investigate the consequences of late *salm* knock-down, we quantified the myofibril and sarcomere morphology from 48 hr onwards. The myofibrils display a fibrillar morphology, confirming that

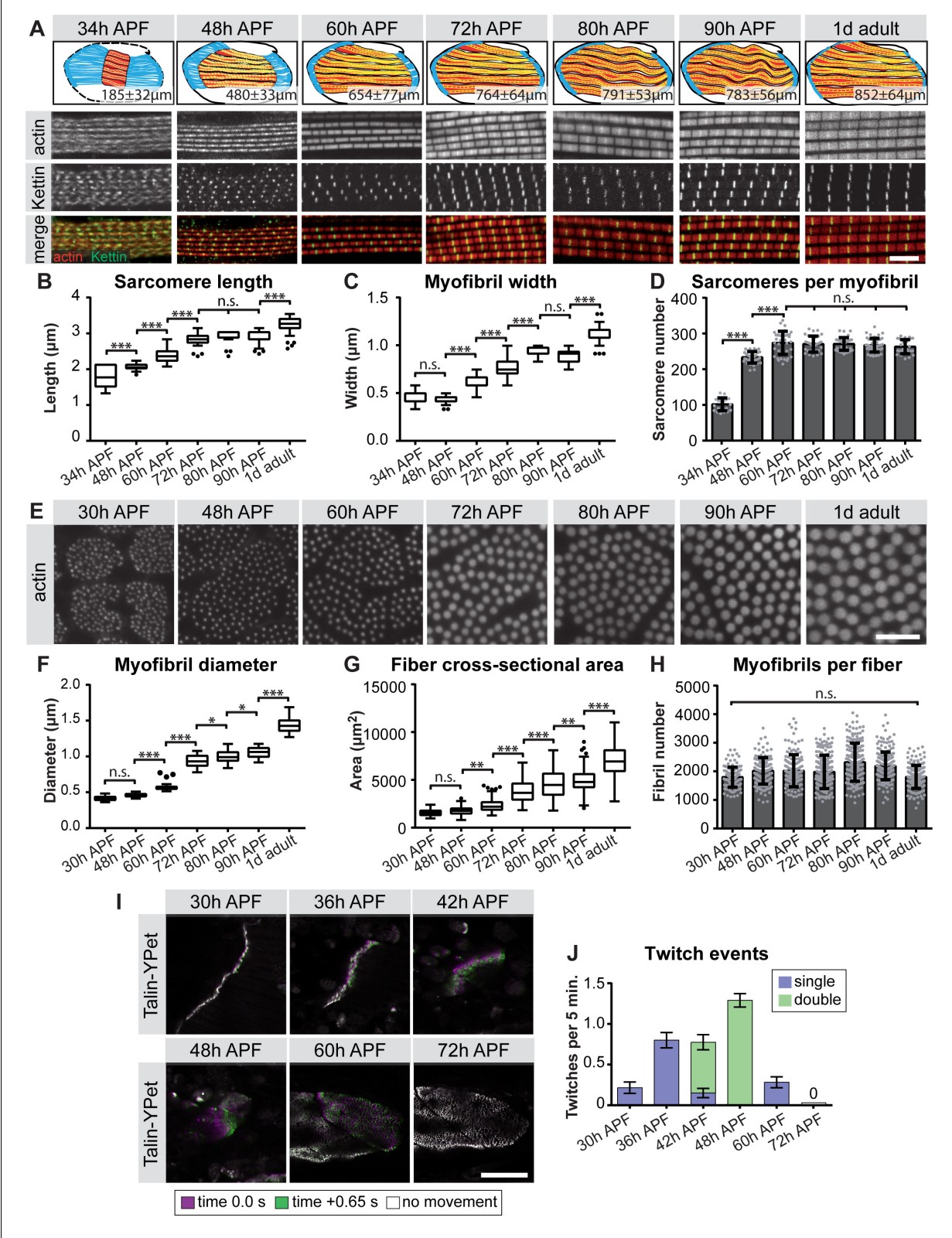

**Figure 5.** Sequential phases of sarcomere morphogenesis in flight muscles. (**A**) Scheme of hemi-thoraces at 34, 48, 60, 72, 80, 90 hr APF and 1 day adults (muscle in red, tendon in blue, sarcomeres in yellow) with indicated muscle fiber length. Myofibrils and sarcomeres at these time points were stained for phalloidin (F-actin, red) and Kettin (Z-disc, green). Scale bar represents 5 μm. (**B,C**) Tukey box and whisker plot of sarcomere length and myofibril width. Box extends from 25 to 75%, line marks median, whiskers extend to 25/75% plus 1.5 times the interquartile range. (**D**) Histogram of
*Figure 5 continued on next page*

Figure 5 continued

sarcomere number per myofibril. Error bars represent SD. Note the sarcomere assembly at 34 hr, followed by sarcomere addition until ~48 hr and sarcomere maturation after ~48 hr APF. (E) Cross-sections of the DLMs at 30, 48, 60, 72, 80, 90 hr APF and 1 day adult. Scale bar represents 5 μm. (F,G) Tukey box and whisker plot of myofibril diameter and myofiber cross-sectional area. Note the lack of growth in diameter or area from 30 hr to 48 hr. (H) Histogram of the number of myofibrils per myofiber. Error bars represent SD. Note that all myofibrils are already present at 30 hr APF. Tukey's multiple comparison p-value<0.05*,. 01**,. 001***, n.s. = not significant. N > 10 for each individual time point. (I) Stills of live movies of DLMs at 30, 36, 42, 48, 60 and 72 hr APF. Scale bar represents 50 μm. For live movies see *Figure 5—video 1*. Stills are a time 0.0 s image (magenta) overlaid with a time +0.65 s image (green), where a perfect overlap (white) shows no movement. (J) Quantification of spontaneous contraction events per fiber per 5 min, with single twitches in blue and double twitches in green. Fibers are first contractile at 30 hr APF, reach peak contractility at 48 hr and stop all spontaneous contraction shortly after 60 hr APF.

DOI: https://doi.org/10.7554/eLife.34058.011

The following video, source data, and figure supplement are available for figure 5:

**Source data 1.** This table includes calculation of the number of sarcomeres per myofibril in wild type flight muscles at the indicated stage.
DOI: https://doi.org/10.7554/eLife.34058.013
**Source data 2.** This table includes the sarcomere length in wild type flight muscles measured at the indicated stage.
DOI: https://doi.org/10.7554/eLife.34058.014
**Source data 3.** This table includes the sarcomere/myofibril width in wild type flight muscles measured at the indicated stage.
DOI: https://doi.org/10.7554/eLife.34058.015
**Source data 4.** This table includes the myofiber cross-sectional area measured in wild type flight muscle fibers or only fibers 3 and four at the indicated stage.
DOI: https://doi.org/10.7554/eLife.34058.016
**Source data 5.** This table includes the number of myofibrils per myofiber calculated for either all wild type flight muscle fibers or only fibers 3 and four at the indicated stage.
DOI: https://doi.org/10.7554/eLife.34058.017
**Source data 6.** This table includes the myofibril diameter measured in wild type flight muscles at the indicated stage.
DOI: https://doi.org/10.7554/eLife.34058.018
**Source data 7.** This table includes the muscle twitch events per wild-type fiber per 5 min recorded at the indicated stage.
DOI: https://doi.org/10.7554/eLife.34058.019
**Figure supplement 1.** Expression and localisation of thin- and thick-filament structural proteins.
DOI: https://doi.org/10.7554/eLife.34058.012
**Figure 5—video 1.** Twitching time-course in developing DLMs.
DOI: https://doi.org/10.7554/eLife.34058.020

the early function of Salm to determine IFM fate was unaffected by our late knock-down. At 72 hr APF and more prominently at 90 hr APF and in adults, *Act88F >> salmIR* myofibrils showed actin accumulations at broadened Z-discs (*Figure 7A–H*), which are often a landmark of nemaline myopathies (*Sevdali et al., 2013*; *Wallgren-Pettersson et al., 2011*). The myofibril width was not significantly different in these myofibrils (*Figure 7I*). However, the sarcomeres of *Act88F >> salmIR* muscles displayed a strong defect in sarcomere length growth after 48 hr APF (*Figure 7J*, *Supplemental File 3*), with sarcomeres only obtaining a length of 2.8 μm in adult flies, demonstrating that Salm in addition to other factors is required for normal sarcomere maturation.

## Salm function contributes to gain of stretch-activation during sarcomere maturation

Given the defects in sarcomere length and sarcomere gene expression in *Act88F >> salmIR* muscles, we explored the function of these abnormal muscle fibers. As expected, *Act88F >> salmIR* flies are flightless (*Figure 7—figure supplement 1A*) and we observed rupturing of the adult muscle fibers within 1d after eclosion (*Figure 7—figure supplement 1B–G*), demonstrating the importance of proper sarcomere maturation to prevent muscle atrophy. Based on our finding that spontaneous flight muscle contractions stop by 72 hr APF, we hypothesized that if Salm truly regulates sarcomere maturation, we may see spontaneous contraction defects during development. At 48 hr APF, *Act88F >> salmIR* fibers twitch, but less often than and without the double twitches observed in control fibers (*Figure 7K,L,O*; *Figure 7—video 1*). Strikingly, at 72 hr APF *salmIR* fibers fail to stop contracting and moreover show frequent and uncoordinated spontaneous contractions in which different myofibril bundles of the same fiber twitch at different times (*Figure 7M–O*, *Figure 7—*

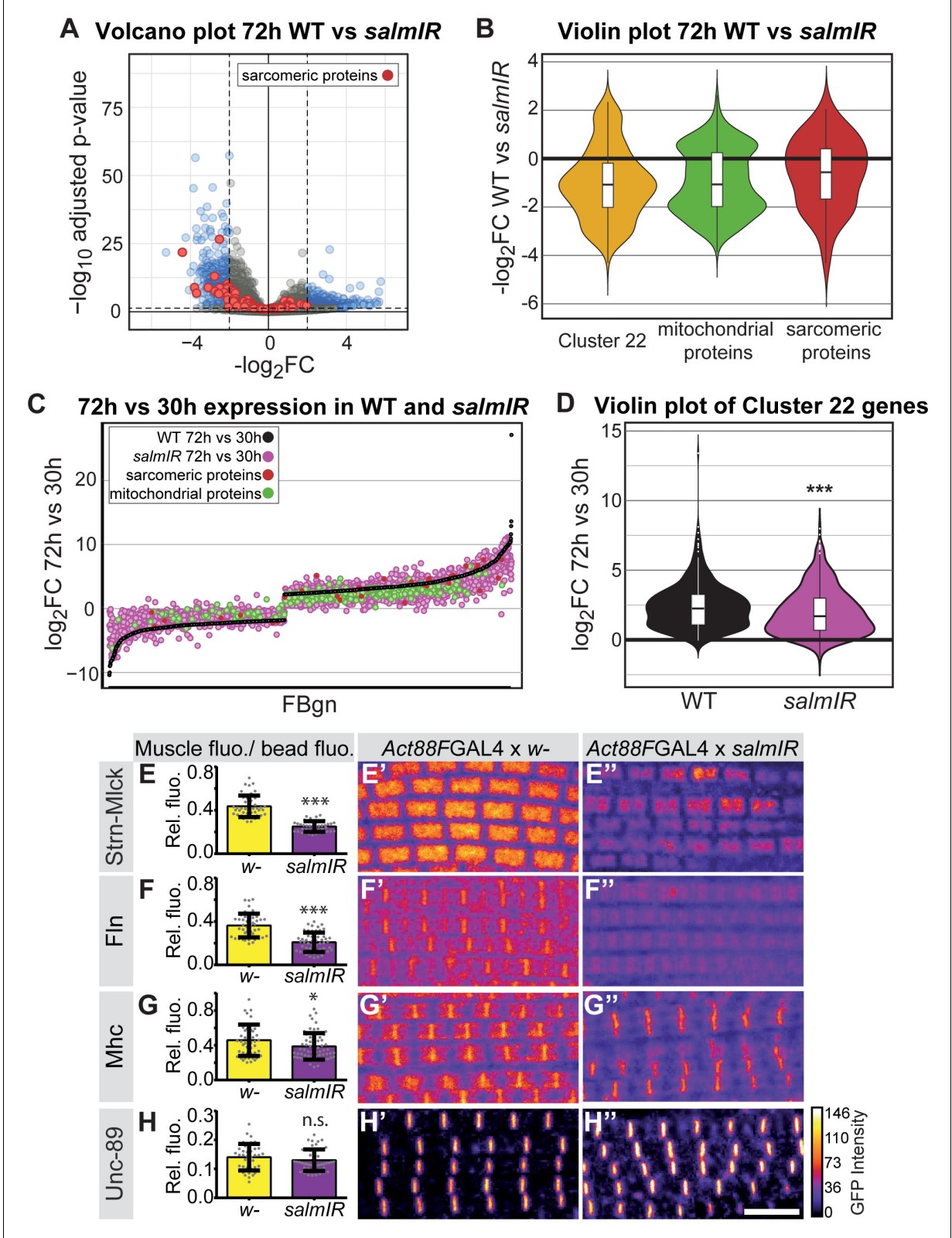

**Figure 6.** *salm* contributes to the transition in gene expression after 30 hr APF. (**A**) Volcano plot of mRNA-Seq comparison of wild-type (WT) versus *salmIR* IFMs at 72 hr APF. Note the significant down-regulation of genes in *salmIR*, especially sarcomeric protein coding genes (red). Significantly differentially expressed (DE) genes (abs(log$_2$ FC)>2, p<0.05) are in blue. (**B**) Violin plot showing down-regulation of sarcomeric proteins (in red), mitochondrial proteins (in green) and members of Mfuzz cluster 22 (in yellow) in *salmIR* compared to wild type at 72 hr APF. Box plots indicate the

*Figure 6 continued on next page*

*Figure 6 continued*

median of the data as well as the first and third quartile in the box, outliers are the dots. (C) WT mRNA-Seq fold change values of all genes significantly DE from 30 hr to 72 hr APF are ordered from lowest to highest (black). The corresponding *salmIR* fold change is shown in yellow. Both sarcomeric (red) and mitochondrial protein coding genes (blue) are less strongly up- or down-regulated in *salmIR* across the 30 hr to 72 hr APF transition. Note that many genes in *salmIR* (yellow dots) are not as strongly induced or even repressed compared to WT (below and above the black WT line, respectively). (D) Violin plot comparing the $\log_2$FC over the 30 hr to 72 hr transition in WT (in black) and *salmIR* IFM (in magenta) for members of Mfuzz Cluster22. Box plots indicate the median of the data as well as the first and third quartile in the box, outliers are the dots. Note the significant decrease in induction to 72 hr APF in the *salmIR* sample. ***Student's t-test p-value<0.0005. (E–H) *salm* is required for the induction of some but not all sarcomeric proteins. *Act88F* >> *salmIR* in the background of GFP-tagged Strn-Mlck-IsoR (E), Fln (F), Mhc (G) and Unc-89 (H). Quantitative changes in live GFP fluorescence at 90 hr APF were measured by quantitative confocal microscopy relative to standard fluorescent beads, revealing significant decreases in induction for Strn-Mlck, Fln and Mhc between wild type control (shown in yellow, *Act88F*-GAL4 crossed to $w^{1118}$) and *Act88F* >> *salmIR* (shown in purple). Scale bar represents 5 μm. Error bars represent SEM, Student's t-test p-value<0.05*, 0.001***, n.s. = not significant. N > 10 for each individual sample. (E'–H'') Intensity-coded GFP fluorescence at 90 hr APF in confocal images of fixed myofibrils.
DOI: https://doi.org/10.7554/eLife.34058.021

The following source data and figure supplements are available for figure 6:

**Source data 1.** This table includes the fiber divided by bead fluorescence intensity measurements for quantification of the indicated fosmid-GFP or UAS-GFP-Gma expression levels at the indicated stage.
DOI: https://doi.org/10.7554/eLife.34058.025

**Source data 2.** This table includes the fiber divided by bead fluorescence intensity measurements for various fosmid-GFPs in the *Act88F* >> *salmIR* or control *Act88F-Gal4 x w-* background at 90 hr APF.
DOI: https://doi.org/10.7554/eLife.34058.026

**Figure supplement 1.** *salm* regulates gene expression during flight muscle development.
DOI: https://doi.org/10.7554/eLife.34058.022

**Figure supplement 2.** *Act88F*-GAL4 driven knock-down of *salm* is efficient.
DOI: https://doi.org/10.7554/eLife.34058.023

**Figure supplement 3.** Expression of sarcomere proteins strongly increases from 30 hr to 72 hr APF.
DOI: https://doi.org/10.7554/eLife.34058.024

---

*video 2*), demonstrating that sarcomere maturation is indeed disrupted, with the likely consequence that myofibrils fail to acquire normal stretch-activation sensitivity.

To directly test the function of a sarcomeric component during the sarcomere maturation phase, we investigated the role of the prominently induced Salm target Strn-Mlck. Strn-Mlck is only expressed after 30 hr APF and is largely incorporated during sarcomere maturation (*Figure 5—figure supplement 1E*, *Figure 6—figure supplement 3E*), and thus is also a bone-fide example of a gene regulated during the transcriptional transition. In *Strn-Mlck* mutants, sarcomere and myofibril morphology, including myofibril width, is initially normal. However, at 80 hr APF the sarcomeres overgrow, consistently reaching lengths of more than 3.5 μm and resulting in slightly longer muscle fibers at 80 hr APF (*Figure 8*). After overgrowing, sarcomeres appear to hyper-contract resulting in short, thick sarcomeres in 1-day-old adults (*Figure 8E,J,K,L*). Like *Act88F* >> *salmIR* flies, *Strn-Mlck* mutant adults are flightless (*Figure 7—figure supplement 1A*) and display ruptured fibers during the first days of life (*Figure 7—figure supplement 1K–M*) (*Spletter et al., 2015*). Together, these data demonstrate that sarcomere maturation must be precisely controlled at the transcriptional level to enable the precise growth of sarcomeres to their final mature size. This ensures the lifelong function of the contractile apparatus of muscle fibers.

## Discussion

### A developmental muscle transcriptomics resource

In this study, we generated a systematic developmental transcriptomics resource from *Drosophila* flight muscle. The resource quantifies the transcriptional dynamics across all the major stages of muscle development over five days, starting with stem cell-like myoblasts and attaching myotubes to fully differentiated, stretch-activatable muscle fibers. We have specifically focused on the transcriptional regulation of sarcomere and myofibril morphogenesis; however, the data we provide cover all other expected dynamics, such as mitochondrial biogenesis, T-tubule morphogenesis, neuromuscular junction formation, tracheal invagination, *etc.* Thus, together with the available systematic

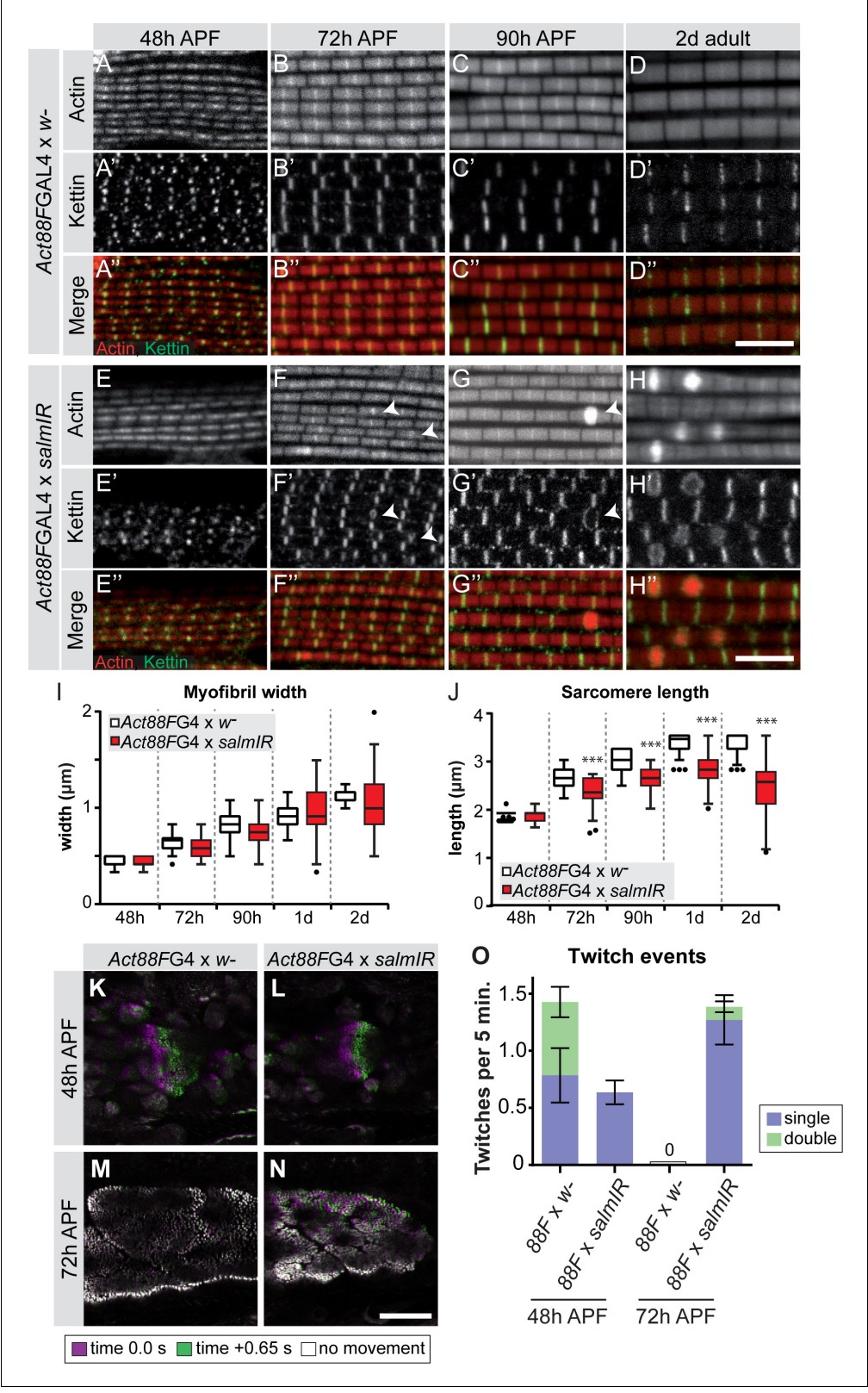

**Figure 7.** *salm* is required for normal sarcomere maturation and function. (A–H) Myofibrils of *Act88F*-GAL4 / + (A–D) or *Act88F* >> *salmIR* (E–H). Note that *salmIR* DLM remains fibrillar and appears normal at 48 hr APF (E), but at 72 hr APF (F) 90 hr APF (G) and 2 day adult (H) Z-discs widen and show actin accumulations (arrowheads). (I,J) Tukey box and whisker plot of myofibril width (I) and sarcomere length (J) in *Act88F*-GAL4 /+ and *Act88F* >> *salmIR* (red). Tukey's multiple comparison p-value<0.001***. N > 10 for each individual time point. Scale bars represent 5 µm. (K–O) Stills

*Figure 7 continued on next page*

*Figure 7 continued*

from live movies of developing DLMs at 48 and 72 hr APF in *Act88F*-GAL4 / + (**K, M**) and *Act88F >> salmIR* (**L, N**). Scale bar represents 50 μm. Coloured as in *Figure 5*. (**O**) Quantification of spontaneous contraction events per fiber per 5 min, with single twitches in blue and double twitches in green. Error bars represent SEM. *salmIR* fibers continue spontaneously contracting at 72 hr APF.

DOI: https://doi.org/10.7554/eLife.34058.027

The following video, source data, and figure supplement are available for figure 7:

**Source data 1.** This table includes the muscle twitch events per *Act88F >> salmIR* fiber per 5 min and the respective controls recorded at the indicated stage.

DOI: https://doi.org/10.7554/eLife.34058.029

**Source data 2.** This table includes the sarcomere length measured in *Act88F >> salmIR* flight muscles and the respective controls at the indicated stage.

DOI: https://doi.org/10.7554/eLife.34058.030

**Source data 3.** This tables includes the sarcomere/myofibril width measured in *Act88F >> salmIR* flight muscles and the respective controls at the indicated stage.

DOI: https://doi.org/10.7554/eLife.34058.031

**Figure supplement 1.** *Act88F >> salmIR* and *Strn-Mlck* mutant flies are flightless and IFM fibers rupture in adult flies.

DOI: https://doi.org/10.7554/eLife.34058.028

**Figure 7—video 1.** Twitching in developing *Act88F*-GAL4 /+ and *Act88F >> salmIR* DLMs at 48 hr APF.

DOI: https://doi.org/10.7554/eLife.34058.032

**Figure 7—video 2.** Twitching in developing *Act88F*-GAL4 /+ and *Act88F >> salmIR* DLMs at 72 hr APF.

DOI: https://doi.org/10.7554/eLife.34058.033

functional data of *Drosophila* muscle development (*Schnorrer et al., 2010*), our data should be a versatile resource for the muscle community. It nicely complements existing systematic data from vertebrate muscle, which thus far are largely restricted to postnatal stages (*Brinegar et al., 2017*; *Drexler et al., 2012*; *Lang et al., 2017*; *Zheng et al., 2009*). Furthermore, *Drosophila* flight muscle contains a single muscle fiber type, in contrast to the mixed fiber types found in mammals (*Schiaffino and Reggiani, 2011*; *Spletter and Schnorrer, 2014*). Hence, in this model the transcriptional dynamics of a single fiber type muscle can be followed with unprecedented precision.

## A transcriptional transition correlating with ordered sarcomere morphogenesis

Earlier work has shown that the flight muscle myotubes first attach to tendon cells and then build-up mechanical tension. This tension triggers the simultaneous assembly of immature myofibrils, converting the myotube to an early myofiber (*Weitkunat et al., 2014*). This suggested a tension-driven self-organisation mechanism of myofibrillogenesis (*Lemke and Schnorrer, 2017*). Here we discovered that myofibrillogenesis is not only regulated mechanically, but to a large extent also transcriptionally. This enabled us to extend our model for ordered myofibrillogenesis also to later developmental stages and to define three sequential although somewhat overlapping phases (*Figure 9*). During the sarcomere self-assembly phase at about 30 hr APF, a large number of genes coding for sarcomeric proteins, including Mhc, Act88F and Unc-89/Obscurin, become up-regulated to enable the self-organization of short, immature sarcomeres within thin, immature myofibrils. Strikingly, all of the about 2000 myofibrils assemble during this phase.

This is followed by a sarcomere addition phase during which a transcriptional transition is initiated and the expression of the sarcomeric proteins increases. Concomitantly, the muscle fibers grow in length by addition of new sarcomeres to all the immature myofibrils, increasing the sarcomere number from about 80 to 230 per fibril at 48 hr APF. These sarcomeres are contractile, but remain short and thin (*Figure 9*).

After the transcriptional transition, myofibrillogenesis enters the final sarcomere maturation phase. Proteins present in immature myofibrils like Mhc, Act88F and Unc-89/Obscurin are expressed to even higher levels, and additional, often flight-muscle specific proteins like Mf, Fln and the titin-related isoform Strn-Mlck, begin to be expressed at high levels and are incorporated into the maturing sarcomeres. This facilitates a dramatic growth of all immature sarcomeres in length and particularly in diameter with all 2000 myofibrils reaching a pseudo-crystalline regularity within about two days of development (*Figure 9*). Importantly, these matured sarcomeres no longer contract

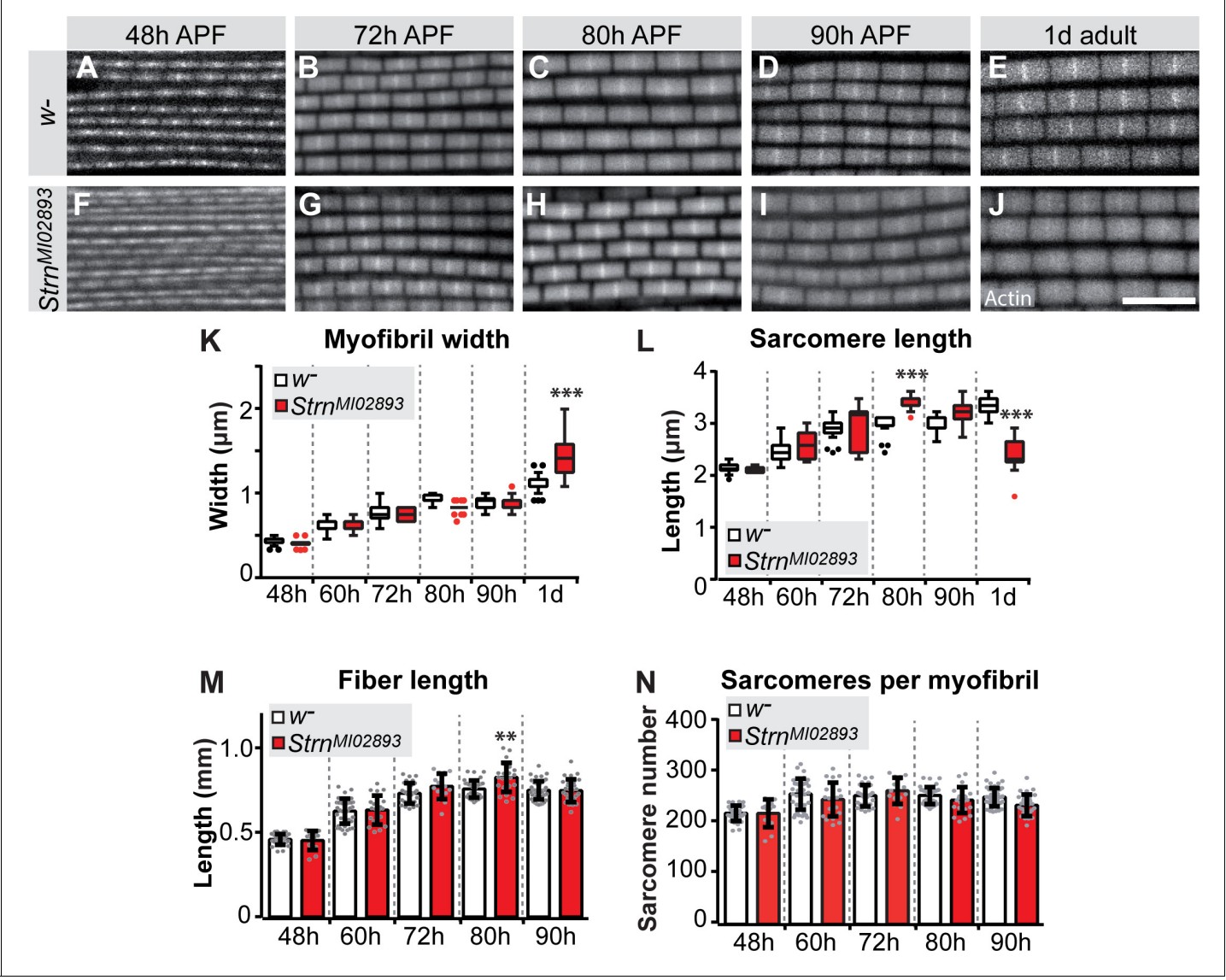

**Figure 8.** Strn-Mlck contributes to sarcomere length regulation during sarcomere maturation. (A–J) Wild type (A–E) and $Strn^{M102893}$ mutant (F–J) sarcomere development at 48, 72, 80, 90 hr APF and 1 day adult. (K–N) Tukey box and whisker plot of myofibril width (K) and sarcomere length (L) in wild type and $Strn^{M102893}$ mutant (red). Tukey's multiple comparison p-value<0.001***. N > 10 for each individual time point. Histogram of fiber length (M) and number of sarcomeres per myofibril (N). Error bars represent SEM. Tukey's multiple comparison p-value<0.01**, N > 10 for each individual time point. Note that a normal number of sarcomeres are formed in $Strn^{M102893}$ mutants, but they grow too long at 80 hr APF and hyper-contract in 1 day adult.

DOI: https://doi.org/10.7554/eLife.34058.034

The following source data is available for figure 8:

**Source data 1.** This table includes the numbers of sarcomeres calculated per *strn-mlck* mutant myofibril and the respective control at the indicated stage.
DOI: https://doi.org/10.7554/eLife.34058.035

**Source data 2.** This table includes the length measurements of the indirect flight muscle fibers at the indicated stage for *strn-mlck* mutants and wild-type controls.
DOI: https://doi.org/10.7554/eLife.34058.036

**Source data 3.** This table includes the calculated numbers of sarcomeres per myofibril at the indicated stage for *strn-mlck* mutants.
DOI: https://doi.org/10.7554/eLife.34058.037

**Source data 4.** This table includes the sarcomere length measured in strn-mlck mutant flight muscles at the indicated stage.
DOI: https://doi.org/10.7554/eLife.34058.038

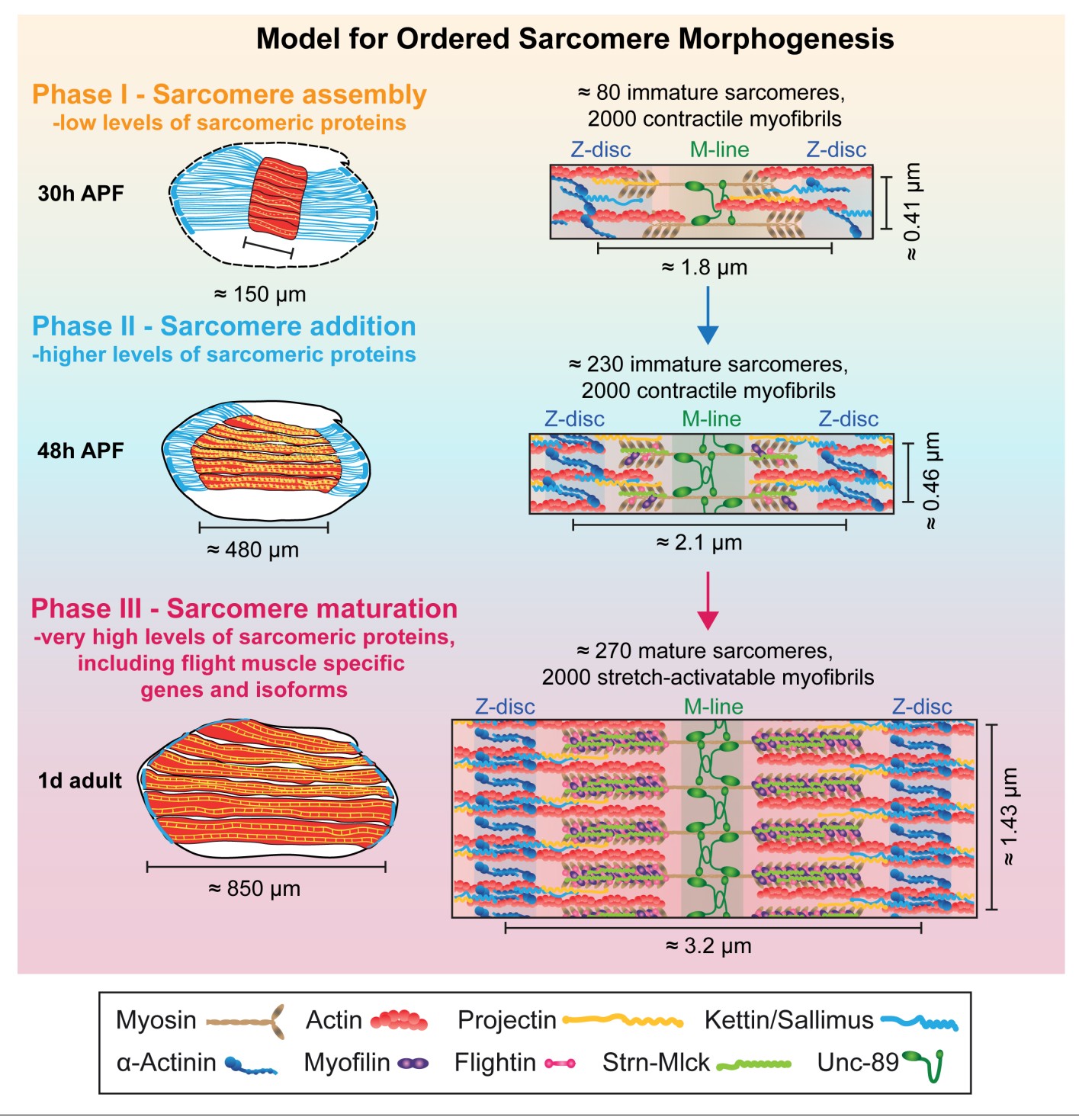

**Figure 9.** Model for ordered sarcomere morphogenesis. Phase 1 - Sarcomere assembly: Sarcomeric proteins are expressed at low levels, enabling the self-assembly of immature sarcomeres and myofibrils at around 30 hr APF. Each of the 2000 myofibrils contains about 80 immature sarcomeres. Phase 2 - Sarcomere addition: Sarcomere protein expression increases and each immature myofibril incorporates many new sarcomeres until about 48 hr APF. These immature sarcomeres contract spontaneously. Phase 3 - Sarcomere maturation: After 48 hr APF, sarcomere protein expression increases even more, including flight muscle specific proteins/isoforms, and all sarcomeres strongly grow in width and length. This enables the flight muscle to gain stretch-activation. Muscles are shown in red, tendons in blue. Structural proteins are illustrated as cartoons and are not drawn to scale. Induction of Actin, Mhc, Myofilin, Flightin, Strn-Mlck and Unc-89/Obscurin is shown here, expression of αActinin, Projectin and Kettin/Sallimus was shown by others (*Bullard et al., 2006*; *Burkart et al., 2007*; *Katzemich et al., 2013*; *Orfanos et al., 2015*; *Weitkunat et al., 2014*).

DOI: https://doi.org/10.7554/eLife.34058.039

spontaneously, likely because they acquired the stretch-activated mechanism of contraction described for mature *Drosophila* flight muscles (*Bullard and Pastore, 2011*; *Josephson, 2006*).

Our ordered sarcomere morphogenesis model is strongly supported by the observation that the number of myofibrils remains largely constant during the entire sarcomere morphogenesis period, suggesting that in flight muscles no new myofibrils are added after the initial assembly of immature myofibrils at about 30 hr APF. As all sarcomeres are present shortly after 48 hr APF, they all need to mature simultaneously to achieve their final pseudo-crystalline regularity.

Our model is further supported by previous studies. We and others found that immature myofibrils have a width of about 0.5 μm (*Weitkunat et al., 2014*), which corresponds to about four thick filaments across each myofibril at the EM level at 42 hr APF (at 22°C) (*Reedy and Beall, 1993*). This 'core' myofibril structure built until 48 hr APF, when most sarcomeres have formed, is expanded dramatically after 48 hr APF, reaching a mature width of 1.5 μm, corresponding to about 35 thick filaments across each myofibril at the EM-level (*Reedy and Beall, 1993*). Recent data showed that the 'core' myofibril structure built until 48 hr APF contains already highly ordered actin filaments, which gain even higher order to reach their pseudo-crystalline regularity at 90 hr APF (*Loison et al., 2018*). In total, each adult myofibril contains around 800 thick filaments per sarcomere (*Gajewski and Schulz, 2010*). The 'core' myofibril structure was also revealed by the preferential recruitment of over-expressed actin isoforms (*Röper et al., 2005*) and more importantly, by selective incorporation of a particular Mhc isoform that is only expressed at mid-stages of flight muscle development (*Orfanos and Sparrow, 2013*). This Mhc isoform expression switch coincides with the global transition in sarcomeric gene expression between the sarcomere assembly and the sarcomere maturation phases that we defined here. It also fits with the recent discovery that the formin family member Fhos is selectively required for actin filament elongation and recruitment of new actin filaments and thus myofibril diameter growth after 48 hr APF (*Shwartz et al., 2016*). Expression of Fhos is also induced after 30 hr APF. Fhos is a member of Mfuzz Cluster 28, another strongly induced cluster, underscoring the general relevance of the transcriptional transition for ordered sarcomerogenesis.

## Regulated active sarcomere contractions

Mature indirect flight muscles employ a stretch-activated mechanism of muscle contraction, thus $Ca^{2+}$ is not sufficient to trigger muscle contractions without additional mechanical stretch (*Bullard and Pastore, 2011*; *Josephson, 2006*). This is different to cross-striated body muscles of flies or mammals that contract synchronously with $Ca^{2+}$ influx. Hence, it is intriguing that immature flight muscle myofibrils do in fact contract spontaneously, with the contraction frequencies and intensities increasing until 48 hr APF. It was recently proposed in *Drosophila* cross-striated abdominal muscles and in developing cross-striated zebrafish muscles that spontaneous contractions are important for the proper formation of the cross-striated pattern (*Mazelet et al., 2016*; *Weitkunat et al., 2017*). A similar role for contractions was found in C2C12 cells by stimulating the contractions optogenetically (*Asano et al., 2015*). This shows that spontaneous contractions are a necessary general feature for the assembly of cross-striated muscle fibers across species.

However, flight muscles are not cross-striated in the classical sense, but have a fibrillar organisation in which each myofibril remains isolated and is not aligned with its neighbouring myofibrils (*Figure 5*) (*Josephson, 2006*; *Schönbauer et al., 2011*). We can only speculate about the mechanism that prevents alignment of the myofibrils in the flight muscles, but it is likely related to their stretch-activated contraction mechanism. This mechanism prevents spontaneous twitching due to increased $Ca^{2+}$ levels, because it additionally requires mechanical activation that can only occur during flight in the adult. Thus, flight muscle sarcomeres not only grow and mature their sarcomere structure, they also gain their stretch-activatability.

## Continuous maintenance of muscle type-specific fate

We identified an important transition in gene expression between the early sarcomere assembly and the late sarcomere maturation phases. Similar large scale transcriptome changes have also been observed during postnatal stages of mouse (*Brinegar et al., 2017*), chicken (*Zheng et al., 2009*) and pig (*Zhao et al., 2015*) skeletal muscle development or during regeneration after injury in fish (*Montfort et al., 2016*) and mouse muscles (*Warren et al., 2007*), indicating that muscle maturation generally correlates with large scale transcriptional changes.

It is well established that general myogenic transcription factors, in particular Mef2, are continuously required in muscles for their normal differentiation (*Sandmann et al., 2006*; *Soler et al., 2012*). Mef2 regulates a suit of sarcomeric proteins in fly, fish and mouse muscle important for correct sarcomere assembly and maturation (*Hinits and Hughes, 2007*; *Kelly et al., 2002*; *Potthoff et al., 2007*; *Stronach et al., 1999*). In *Drosophila*, Mef2 cooperates with tissue-specific factors, such as CF2, to induce and fine-tune expression of structural genes (*Gajewski and Schulz, 2010*; *García-Zaragoza et al., 2008*; *Tanaka et al., 2008*). General transcriptional regulators, such as E2F, further contribute to high levels of muscle gene expression during myofibrillogenesis, in part through regulation of Mef2 itself (*Zappia and Frolov, 2016*). However, it is less clear if muscle type-specific identity genes are continuously required to execute muscle type-specific fate. Spalt major (Salm) is expressed after myoblast fusion in flight muscle myotubes and is required for all flight muscle type-specific gene expression: in its absence the fibrillar flight muscle is converted to tubular cross-striated muscle (*Schönbauer et al., 2011*; *Spletter et al., 2015*). Here we demonstrated that Salm is continuously required for correct sarcomere morphogenesis, as late *salm* knock-down leads to defects in sarcomere growth during the late sarcomere maturation phase, causing severe muscle atrophy in adults. It might do so by modifying the cooperation between Mef2 and E2F or by changing chromatin states, as vertebrate spalt homologs are recently discussed as epigenetic regulators (*Yang, 2018*; *Zhang et al., 2015*). However, Salm cannot be solely responsible for the transcriptional transition after 30 hr APF, as the transition still partially occurs in its absence.

## Ordered sarcomere morphogenesis – a general mechanism?

Here we defined ordered phases of sarcomere morphogenesis in *Drosophila* flight muscles. Is this a general concept for sarcomere morphogenesis? Reviewing the literature, one finds that in other *Drosophila* muscle types which display a tubular cross-striated myofibril organisation, such as the fly abdominal muscles, the striated sarcomeres also first assemble and then grow in length (*Perez-Pérez-Moreno et al., 2014*; *Weitkunat et al., 2017*), suggesting a conserved mechanism. In developing zebrafish skeletal muscles, young myofibers present in younger somites show a short sarcomere length of about 1.2 μm, which increases to about 2.3 μm when somites and muscle fibers mature (*Sanger et al., 2017*; *Sanger et al., 2009*). Interestingly, sarcomere length as well as thick filament length increase simultaneously during fish muscle maturation, indicating that as in flight muscles, the length of all sarcomeres in one large muscle fiber is homogenous at a given time (*Sanger et al., 2009*). Similar results were obtained in mouse cardiomyocytes measuring myosin filament length at young (two somite) and older (13 somite) stages (*Du et al., 2008*) and even in human cardiomyocytes, in which myofibrils increase nearly threefold in width and become notably more organized and contractile from 52 to 127 days of gestation (*Racca et al., 2016*). These data suggest that sarcomeres generally may go through a series of ordered but overlapping developmental phases.

Interestingly, these changes in sarcomere morphology correlate with a switch in myosin heavy chain isoform expression changing from embryonic to neonatal to adult during skeletal muscle development (*Schiaffino et al., 2015*). Importantly, mutations in embryonic myosin (MYH3 in humans) result in severe congenital disorders characterised by multiple facial and limb contractures (*Toydemir et al., 2006*). As a similar ordered expression of myosin isoforms is also found during muscle regeneration after injury in adults (*Ciciliot and Schiaffino, 2010*; *Schiaffino et al., 2015*), we hypothesize that our sequential sarcomere morphogenesis model may also be applicable to vertebrate skeletal and possibly heart muscles. It will be a future challenge to identify possible feedback mechanisms that indicate the successful end of the sarcomere assembly phase or a possible re-entry into the sarcomere assembly phase during muscle regeneration or exercise induced muscle fiber growth. It is enticing to speculate that the assembling cytoskeleton itself would measure its assembly status and provide a mechanical feedback signal to modify the activity of muscle-specific transcription factors.

## Materials and methods

**Key resources table**

*Continued on next page*

*Continued*

| Reagent type (species) or resource | Designation | Source or reference | Identifiers | Additional information |
|---|---|---|---|---|
| Reagent type (species) or resource | Designation | Source or reference | Identifiers | Additional information |
| Gene (*Drosophila melanogaster*) | spalt major; salm | NA | FLYB:FBgn0261648 | |
| Gene (*D. melanogaster*) | Stretchin-Mlck; Strn-Mlck | NA | FLYB:FBgn0265045 | |
| Genetic reagent (*D. melanogaster*) | w[1118] | Bloomington | BDSC:3605; FLYB:FBst0003605; RRID:BDSC_3605 | |
| Genetic reagent (*D. melanogaster*) | salmIR | PMID: 22094701 | VDRC:13302; FLYB:FBst0450930 | Flybase symbol: VDRC:v13302 |
| Genetic reagent (*D. melanogaster*) | KK101052 | PMID: 17625558 | VDRC:101052 | |
| Genetic reagent (*D. melanogaster*) | Act88F-GAL4 | PMID: 22008792 | | Source: Richard Cripps |
| Genetic reagent (*D. melanogaster*) | Strn-Mlck-MiMIC | Bloomington | FLYB:FBal0264439 | Flybase symbol:Strn-MlckMI02893 |
| Genetic reagent (*D. melanogaster*) | Strn-Mlck-IR | PMID: 21460824 | BDSC:31891; FLYB:FBti0130299; RRID:BDSC_31891 | Flybase symbol:P{TRiP.JF02170}attP2 |
| Genetic reagent (*D. melanogaster*) | Strn-Mlck-GFP, Isoform R | PMID: 25532219 | | Symbol: Strn4; |
| Genetic reagent (*D. melanogaster*) | Mhc-GFP | PMID: 26896675 | VDRC:318471 | Symbol: fTRG500; |
| Genetic reagent (*D. melanogaster*) | Mf-GFP | PMID: 26896675 | VDRC:318132 | Symbol: fTRG501; |
| Genetic reagent (*D. melanogaster*) | rhea-GFP | PMID: 26896675 | VDRC:318486 | Symbol: fTRG587; |
| Genetic reagent (*D. melanogaster*) | Fln-GFP | PMID: 26896675 | VDRC:318238 | Symbol: fTRG876; |
| Genetic reagent (*D. melanogaster*) | mys-GFP | PMID: 26896675 | VDRC:318285 | Symbol: fTRG932; |
| Genetic reagent (*D. melanogaster*) | βTub60D-GFP | PMID: 26896675 | VDRC:318309 | Symbol: fTRG958; |
| Genetic reagent (*D. melanogaster*) | unc-89-GFP | PMID: 26896675 | VDRC:318326 | Symbol: fTRG1046; |
| Genetic reagent (*D. melanogaster*) | Act88F-GFP | PMID: 26896675 | VDRC:318362 | Symbol:fTRG10028; |
| Genetic reagent (*D. melanogaster*) | Him-nuc-eGFP | PMID: 19324085 | | Source: Michael V. Taylor |
| Genetic reagent (*D. melanogaster*) | Him-GAL4 | this paper | | |
| Genetic reagent (*D. melanogaster*) | UAS-BBM | PMID: 22446736 | | |
| Genetic reagent (*D. melanogaster*) | Him-GFP-Gma | this paper | | |
| Genetic reagent (*D. melanogaster*) | rhea-YPet | this paper | | |
| Genetic reagent (*D. melanogaster*) | kon-GFP | this paper | | |
| Genetic reagent (*D. melanogaster*) | Mef2-GAL4 | Bloomington | BDSC:27390; RRID:BDSC_27390 | |

*Continued on next page*

*Continued*

| Reagent type (species) or resource | Designation | Source or reference | Identifiers | Additional information |
|---|---|---|---|---|
| Genetic reagent (*D. melanogaster*) | UAS-GFP-Gma | PMID: 12324971 | | Source:Don Kiehart; Description |
| Antibody | guinea pig anti-Shot | PMID: 9832554 | | (1:500); Source: Talila Volk |
| Antibody | rat anti-Kettin (MAC155/Klg16) | Babraham Bioscience Technologies | Babraham: MAC_155(P6689) | (1:50) |
| Antibody | rabbit anti-GFP (ab290) | Abcam | Abcam:ab290 | (1:1000) |
| Antibody | rat anti-Bruno | PMID: 12591598 | | (1:500); Source: Anne Ephrussi |
| Antibody | rabbit anti-Salm | PMID: 7905822 | | (1:50); Source: Reinhard Schuh |
| Antibody | mouse anti-βPS-integrin (CF.6G11) | Developmental Studies Hybridoma Bank | DSHB:CF.6G11 | (1:500) |
| Antibody | rabbit anti-Twi | PMID: 2688897 | | (1:1000); Source: Siegfried Roth |
| Antibody | rabbit anti-Fln | PMID: 11134077 | | (1:50); Source: Jim Vigoreaux |
| Commercial assay or kit | fluorescent beads | ThermoFisher (Molecular Probes) | | OrderID: InSpeckTM Green Kit I-7219 |
| Commercial assay or kit | Dynabeads | Invitrogen | | OrderID: #610.06 |
| Commercial assay or kit | Superscript III First-Strand Synthesis System | Invitrogen | | OrderID: #18080–051 |
| Chemical compound, drug | Fluoroshield with DAPI | Sigma | | OrderID: #F6057 |
| Chemical compound, drug | Vectashield with DAPI | Biozol | | OrderID: VEC-H-1200 |
| Chemical compound, drug | Tissue-Tek O.C.T. | Weckert Labotechnik | | OrderID: 4583; Sakura Finetek |
| Chemical compound, drug | TriPure reagent | Roche | | OrderID: #11667157001 |
| Software, algorithm | Fiji (Image J) | PMID: 22743772 | | |
| Software, algorithm | MyofibrilJ | this paper | 1dbb0d4 | Source: https://imagej.net/MyofibrilJ |
| Software, algorithm | STAR | PMID: 23104886 | | |
| Software, algorithm | SAMtools | PMID: 19505943 | | |
| Software, algorithm | featureCounts | PMID: 24227677 | | |
| Software, algorithm | DESeq2 | PMID: 25516281 | | |
| Software, algorithm | R | R Project for Statistical Computing | RRID:SCR_001905 | |
| Software, algorithm | ComplexHeatmap | PMID: 27207943 | | |
| Software, algorithm | corrplot | GitHub | | Source: https://github.com/taiyun/corrplot |
| Software, algorithm | VennDiagram | PMID: 21269502 | | |
| Software, algorithm | plyr | DOI: 10.18637/jss.v040.i01 | | |
| Software, algorithm | reshape2 | DOI: 10.18637/jss.v021.i12 | | |
| Software, algorithm | ggplot2 | ISBN:978-0-387-98140-6 | | |
| Software, algorithm | RColorBrewer | Author: Erich Neuwirth; | | Source: https://cran.r-project.org/web/packages/RColorBrewer/index.html |
| Software, algorithm | Mfuzz | PMID: 16078370 | | |
| Software, algorithm | GO-Elite | PMID: 22743224 | | |

## Fly strains

Fly stocks were maintained using standard culture conditions. Characterization of normal IFM sarcomere and fiber growth was performed in $w^{1118}$ grown at 27°C. *salm* RNAi was performed with previously characterized GD3029 (referred to as *salmIR*) and KK181052 (*Schönbauer et al., 2011*) from VDRC (http://stockcenter.vdrc.at) (*Dietzl et al., 2007*) at 25°C using *Act88F*-GAL4 to induce knockdown after 24 hr APF. *Act88F*-GAL4 x $w^{1118}$ served as control. The *Strn-Mlck-MiMIC* insertion MI02893 into IFM-specific IsoR (Bloomington stock 37038) and TRiP hairpin JF02170 were obtained from Bloomington (*Ni et al., 2011*). The *salm*-EGFP line was used to sort wing discs (*Marty et al., 2014*). Tagged genomic fosmid reporter fly lines include *strn4* (*Strn-Mlck-GFP*, Isoform R) (*Spletter et al., 2015*), fTRG500 (*Mhc-GFP*, Isoforms K, L, M), fTRG501 (*Mf-GFP*, Isoforms A, G, N), fTRG587 (*Rhea-GFP*, Isoforms B, E, F, G), fTRG876 (*Fln-GFP*), fTRG932 (*mys-GFP*), fTRG958 (*βTub60D-GFP*), fTRG1046 (*unc-89-GFP*), and fTRG10028 (*Act88F-GFP*) (*Sarov et al., 2016*).

To label myoblasts, we utilized the enhancer for *Holes-in-muscle* (*Him*), which is expressed in dividing myoblasts and promotes the progenitor fate. *Him-nuc-eGFP* flies were a gift of M. Taylor (*Soler and Taylor, 2009*). *Him*-GAL4 flies were created by cloning an EcoRI to SacII fragment of the *Him* enhancer (*Liotta et al., 2007*) upstream of GAL4 into pStinger. UAS-BBM (UAS-palmCherry) (*Förster and Luschnig, 2012*) was driven with *Him*-GAL4 to label myoblasts. *Him*-GFP-Gma flies were created by PCR amplifying GFP-Gma with AscI and PacI overhangs and then cloning downstream of the Him enhancer in pStinger to generate a gypsy insulator-$Him^{enh}$-Gma-GFP-SV40-gypsy insulator cassette.

The *rhea-YPet* line used to label muscle ends for live imaging of twitch events was generated by CRISPR-mediated gene editing at the endogenous locus (S.B.L and F.S., details will be published elsewhere). The *kon-GFP* line was generated by inserting GFP into the *kon* locus after its transmembrane domain using the genomic fosmid FlyFos021621, which was integrated using Φ-C31 into VK00033 (I. Ferreira and F.S., details will be published elsewhere).

## Flight tests

Flight tests were performed as previously described (*Schnorrer et al., 2010*). *Act88F*-GAL4 crosses were kept at 25°C, as higher temperatures negatively impacted flight ability, because of the very high GAL4 expression levels in this strain. Adult males were collected on $CO_2$ and recovered at least 24 hr at 25°C before testing. Flies were introduced into the top of a 1 m long cylinder divided into five zones. Those that landed in the top two zones were considered 'normal fliers', those in the next two zones 'weak fliers' and those that fell to the bottom of the cylinder 'flightless'.

## Immuno-staining

Wing-discs were dissected from 3rd instar wandering larvae in 1x PBS and fixed in 4% PFA in PBS-T. Discs were stained as described below for anti-GFP. Adult and pupal flight muscles were dissected and stained as previously described (*Weitkunat and Schnorrer, 2014*). Briefly, early pupae (16–60 hr APF) were freed from the pupal case, fixed for 20 min. in 4% PFA in relaxing solution and washed in 0.5% PBS-Triton-X100 (PBS-T). 72 hr APF and older samples were cut sagittally with a microtome blade. All samples were blocked for at least 1 hr at RT in 5% normal goat serum in PBS-T and stained with primary antibodies overnight at 4°C. Primary antibodies include: guinea pig anti-Shot 1:500 (gift of T. Volk), rat anti-Kettin 1:50 (MAC155/Klg16, Babraham Institute), rabbit anti-GFP 1:1000 (ab290, Abcam), rat anti-Bruno 1:500 (*Filardo and Ephrussi, 2003*), rabbit anti-Salm 1:50 (*Kühnlein et al., 1994*), mouse anti-βPS-integrin 1:500 (CF.6G11, DSHB), rabbit anti-Twi 1:1000 (gift of Siegfried Roth) and rabbit anti-Fln 1:50 (*Reedy et al., 2000*) (gift of Jim Vigoreaux). Samples were washed three times in 0.5% PBS-T and incubated overnight at 4°C with secondary conjugated antibodies (1:500) from Invitrogen (Molecular Probes) including: Alexa488 goat anti-guinea pig IgG, Alexa488 donkey anti-rat IgG, Alexa488 goat anti-mouse IgG, Alexa488 goat anti-rabbit IgG, rhodamine-phalloidin, Alexa568 goat anti-rabbit IgG and Alexa633 goat anti-mouse IgG. Samples were washed three times in 0.5% PBS-T and mounted in Vectashield containing DAPI.

## Cryosections

Head, wings and abdomen were removed from one day old $w^{1118}$ flies and thoraxes were fixed overnight at 4°C in 4% PFA. For 30–90 hr APF samples, pupae were freed from the pupal case, poked 3–

5 times with an insect pin in the abdomen and fixed overnight at 4°C in 4% PFA. Thoraxes or pupae were then sunk in 30% sucrose in 0.5% PBS-T overnight at 4°C on a nutator. Thoraxes or pupae were embedded in Tissue-Tek O.C.T. (Sakura Finetek) in plastic moulds (#4566, Sakura Finetek) and frozen on dry ice. Blocks were sectioned at 30 μm on a cryostat (Microm vacutome). Sections were collected on glass slides coated with 1% gelatin +0.44 μM chromium potassium sulfate dodecahydrate to facilitate tissue adherence. Slides were post-fixed for 1 min. in 4% PFA in 0.5% PBS-T at RT, washed in 0.5% PBS-T, incubated with rhodamine-phalloidin for 2 hr at RT, washed three times in 0.5% PBS-T and mounted in Fluoroshield with DAPI (#F6057, Sigma).

## Microscopy and image analysis

Images were acquired with a Zeiss LSM 780 confocal microscope equipped with an α Plan-APO-CHROMAT 100x oil immersion objective lens (NA 1.46). To compare if indicator protein expression replicates the mRNA-Seq expression dynamics, we imaged three time points from each expression profile with the same confocal settings. Laser gain and pinhole settings were set on the brightest sample and reused on remaining time points in the same imaging session. All samples were additionally stained with the same antibody mix on the same day and if possible in the same tube. Images were processed with Fiji (*Schindelin et al., 2012*) and Photoshop, and displayed using the 'Fire' look-up table.

Fiber length and fiber cross-sectional area were measured with freehand drawing tools in Fiji based on rhodamine-phalloidin staining. Sarcomere length, myofibril width, and myofibril diameter were measured automatically using a custom Fiji plug-in, MyofibrilJ, available from https://imagej.net/MyofibrilJ and code from https://github.com/giocard/MyofibrilJ (*Cardone, 2018*; copy archived at https://github.com/elifesciences-publications/MyofibrilJ). All measurements are based on rhodamine-phalloidin staining, except 34 hr APF sarcomere lengths, which are based on both rhodamine-phalloidin and Unc-89-GFP staining. 'Sarcomeres per fibril' was calculated as average individual fiber length divided by sarcomere length for fiber 3 or 4. 'Fibrils per fiber' was calculated as average number of fibrils per unit area multiplied by individual fiber cross sectional area.

For determining myofibril diameter, samples were imaged using a 3x optical zoom (50 nm pixel size). At least 20 cross-section images from different fibers for >10 flies were acquired for each time point. The number of fibrils per section and fibril diameter were determined with the tool 'analyze myofibrils crosswise' from MyofibrilJ (https://imagej.net/MyofibrilJ). In this tool, an initial estimate of the diameter is obtained by finding the first minimum in the radial average profile of the autocorrelation (*Goodman, 1996*) of the image. This estimate is used to calibrate the optimal crop area around all the cross-sections in the image, their position previously detected by finding the local intensity peaks. All of the detected cross-sections are then combined to obtain a noise-free average representation of the fibril section. Finally, the diameter is calculated by examining the radial profile of the average and measuring the full width where the intensity is 26% of the maximum range.

For determining sarcomere length and myofibril width, for each experiment between 10 and 25 images were acquired from more than 10 individual flies. From each image, nine non-overlapping regions of interest were selected, which were rotated to orient fibrils horizontally, when necessary. The tool 'analyze myofibrils lengthwise' from MyofibrilJ reports the sarcomere length (indicated as repeat) and myofibril width (indicated as thickness). Because of the periodic nature of sarcomere organization, their length is estimated by means of Fourier analysis, identifying the position of the peaks on the horizontal axis of the Fourier transformed imaged. Quality of the estimate was evaluated by visual inspection of the Fourier transformed image, overlaid with the peaks detected, as generated by the plug-in. Myofibrils width is estimated from the position of the first minimum in the vertical intensity profile of the autocorrelation of the image.

Live imaging of developmental spontaneous contractions was performed on a Leica SP5 confocal microscope. Prior to imaging, a window was cut in the pupal case, and pupae were mounted in slotted slides as previously described (*Lemke and Schnorrer, 2018*; *Weitkunat and Schnorrer, 2014*). At the specified developmental time point, IFMs were recorded every 0.65 s for 5 min. General movement within the thorax was distinguished from IFM-specific contraction, and each sample was scored for the number of single or double contractions observed per 5 min time window. Data were recorded in Excel and ANOVA was performed in GraphPad Prism to determine significant differences. Movies were assembled in Fiji (Image J), cropped and edited for length to highlight a selected twitch event.

Quantitative imaging of fosmid reporter intensity was performed at 30 hr APF, 48 hr APF, 72 hr APF, 90 hr APF and in 1 day adult in live IFM by normalizing to fluorescent beads (ThermoFisher (Molecular Probes), InSpeck Green Kit I-7219). IFMs were dissected from five flies, mounted with fluorescent microspheres (0.3% or 1% relative intensity, depending on the reporter intensity) in the supplied mounting medium and immediately imaged (within 20 min). Intensity measurements were obtained at 40x for at least 10 flies in regions where both IFM and at least three beads were visible. Control *Act88F*-GAL4;; *fosmid-GFP* x *w*[1118] and *Act88F*-GAL4;; *fosmid-GFP* x *salmIR* (fosmids used included *Strn-Mlck-GFP*, *Mhc-GFP*, *Fln-GFP*, and *Unc-89-GFP*) were imaged in the same imaging session. Relative fluorescence fiber to beads was calculated for each image in Fiji by averaging intensity for three fiber ROIs and three bead ROIs. Data were recorded in Excel and Student's t-test for significance and plotting were performed in GraphPad Prism.

## mRNA-Seq

We previously published mRNA-Seq analysis of dissected IFMs from *Mef2*-GAL4, UAS-GFP-Gma x *w*[1118] at 30 hr APF, 72 hr APF and 1d adult, and *Mef2-GAL4, UAS-GFP-Gma* x *salmIR* in 1d adult (*Spletter et al., 2015*). We expanded this analysis in the present study to include myoblasts from third instar larval wing discs (see below) and dissected IFMs from *Mef2*-GAL4, UAS-GFP-Gma x *w*[1118] at 16, 24, 30, 48, 72, 90 hr APF and from 1 day adults as well as IFMs from *Mef2*-GAL4, UAS-GFP-Gma x *salmIR* flies at 24, 30, 72 hr APF and from 1 day adults. IFMs were dissected from groups of 15 flies in 30 min to minimize changes to the transcriptome, spun down in PBS for 5 min at 7500 rpm and immediately frozen in 100 µl TriPure reagent (#11667157001, Roche) on dry ice. RNA was isolated after combining IFMs from 150 to 200 flies, with biological duplicates or triplicates for each time point.

Poly(A)+mRNA was purified using Dynabeads (#610.06, Invitrogen) and integrity was verified on a Bioanalyzer. mRNA was then fragmented by heating to 94°C for 210 s in fragmentation buffer (40 mM TrisOAc, 100 mM KOAc, 30 mM MgOAc$_2$). First-strand cDNA synthesis was performed with the Superscript III First-Strand Synthesis System (#18080–051, Invitrogen) using random hexamers. The second strand was synthesized with dUTP and submitted to the Vienna Biocenter Core Facilities (VBCF, http://www.vbcf.ac.at) for stranded library preparation according to standard Illumina protocols and sequenced as SR100 on an Illumina HiSeq2500. Libraries were multiplexed two to four per lane using TrueSeq adaptors.

## Wing disc sorting and myoblast isolation

To perform mRNA-Seq on fusion competent myoblasts that will form the IFMs, we first dissected wing discs from wandering third instar larvae and manually cut the hinge away from the wing pouch. mRNA was isolated in TriPure reagent and sequenced as described above. We estimate this sample (Myo1) is ~50% myoblast, as the myoblasts form a nearly uniform layer over the underlying epithelial monolayer. To obtain a purer myoblast sample, we performed large-scale imaginal disc sorting followed by dissociation. We used particle sorting to isolate imaginal discs from *Him*-GAL4, UAS-BBM (UAS-palmCherry); *salm*-EGFP flies based on the green fluorescent signal. 10–12 ml of larvae in PBS were disrupted using a GentleMACS mixer (Miltenyl Biotec) and discs were collected through a mesh sieve (#0278 in, 25 opening, 710 µm). Fat was removed by centrifugation for 10 min. at 1000 rpm at 4°C, discs were rinsed in PBS and then re-suspended in HBSS. Discs were further purified on a Ficoll gradient (25%:16%). Discs were then sorted on a Large Particle Flow Cytometer (BioSorter (FOCA1000), Union Biometrica, Inc.), obtaining 600–1000 discs per sample. Discs were spun for 5 min at 600 rcf in a Teflon Eppendorf tube and then re-suspended in the dissociation mixture (200 µl of 10x Trypsin, 200 µl HBSS, 50 µl collagenase (10 mg/mL), 50 µl dispase (10 mg/ml)). The tube was incubated for 10 min. at RT and then transferred to a thermal shaker for 30 min. at 25°C at 650 rpm. Myoblasts were filtered through a 35 µm tube-cap filter and spun at 600 rcf for 5 min. to pellet the cells. Cells were re-suspended in HBSS for evaluation or frozen in TriPure reagent for RNA extraction. We obtained samples with ~90% purity based on counting the number of red fluorescent cells/non fluorescent+green fluorescent cells in three slide regions. mRNA was isolated in TriPure reagent and sequenced as described above, generating the Myo2 and Myo3 samples.

## Analysis of RNA-Seq data

FASTA files were de-multiplexed and base called using Illumina software. Reads were trimmed using the FASTX-toolkit. Sequences were mapped using STAR (*Dobin et al., 2013*) to the *Drosophila* genome (BDGP6.80 from ENSEMBL). Mapped reads were sorted and indexed using SAMtools (*Li et al., 2009*), and then bam files were converted to bigwig files. Libraries were normalized based on library size and read-counts uploaded to the UCSC Browser for visualization (http://genome.ucsc. edu/cgi-bin/hgTracks?hgS_doOtherUser=submit&hgS_otherUserName=Ayeroslaviz&hgS_otherU-serSessionName=IFMTP.leg.TCpaperHub1; http://genome.ucsc.edu/cgi-bin/hgTracks?hgS_doO-therUser=submit&hgS_otherUserName=Ayeroslaviz&hgS_otherUserSessionName=AretSalmIFMTP. TCpaperHub2).

Mapped sequences were run through featureCounts (*Liao et al., 2014*) and differential expression analysis was performed on the raw counts using DESeq2 (*Love et al., 2014*). We performed all pairwise comparisons across the time-course as well as between wild-type and *salmIR* samples (*Supplementary file 1* and *4*). All original and processed data can be found as supplemental data or in the Gene Expression Omnibus submission (accession number GSE107247). R packages employed in the analysis include ComplexHeatmap (*Gu et al., 2016*), CorrPlot (https://github.com/taiyun/corr-plot), VennDiagram (*Chen and Boutros, 2011*), plyr (*Wickham, 2011*), reshape2 (*Wickham, 2007*), ggplot2 (*Wickham, 2009*) and RColorBrewer (*Neuwirth, 2015*).

Genome-wide soft clustering was performed in R with Mfuzz (*Futschik and Carlisle, 2005*), using the DESeq2 normalized count values. We filtered the dataset to include all genes expressed at one time point or more, defining expression as >100 counts after normalization. We then set all count values < 100 to 0, to remove noise below the expression threshold. DESeq2 normalized data was standardized in Mfuzz to have a mean value of zero and a standard deviation of one, to remove the influence of expression magnitude and focus on the expression dynamics. We tested 'k' ranging from 10 to 256. We then performed consecutive rounds of clustering to obtain three independent replicates with similar numbers of iterations, ultimately selecting a final k = 40 clusters with iterations equal to 975, 1064 and 1118. We calculated a 'stability score' for each cluster by calculating how many genes are found in the same cluster in each run (*Supplementary file 1*). Figures are from the 1064 iterations dataset. Mfuzz cluster core expression profiles were calculated as the average standard-normal expression of all genes with a membership value greater than or equal to 0.8, and then core profiles were clustered in R using Euclidean distance and complete linkage.

Enrichment analysis was performed with GO-Elite (*Zambon et al., 2012*) using available Gene Ontology terms for *Drosophila*. We additionally defined user provided gene lists for transcription factors, RNA binding proteins, microtubule associated proteins, sarcomeric proteins, genes with an RNAi phenotype in muscle (*Schnorrer et al., 2010*), mitochondrial genes (http://mitoXplorer.bio-chem.mpg.de) and *salm* core fibrillar genes (*Spletter et al., 2015*). Full results and gene lists are available in *Supplementary file 2*. These user-supplied lists allowed us to define more complete gene sets relevant to a particular process or with a specific localization than available in existing GO terms. Analysis was performed with 5000 iterations to generate reliable significance values.

## Data availability

Processed data from DESeq2, Mfuzz and GO-Elite are available in *Supplementary file 1*, *2*, *4*. mRNA-Seq data are publicly available from NCBI's Gene Expression Omnibus (GEO) under accession number GSE107247. mRNA-Seq read counts are further publicly accessible as track hubs in the UCSC Genome Browser at the following links: [1] (wild-type IFM time course) http://genome. ucsc.edu/cgi-bin/hgTracks?hgS_doOtherUser=submit&hgS_otherUserName=Ayeroslaviz&hgS_oth-erUserSessionName=IFMTP.leg.TCpaperHub1 and [2] (salm timecourse) http://genome.ucsc.edu/ cgi-bin/hgTracks?hgS_doOtherUser=submit&hgS_otherUserName=Ayeroslaviz&hgS_otherUserSes-sionName=AretSalmIFMTP.TCpaperHub2. Fiji scripts for analysis of sarcomere length, myofibril width and myofibril diameter are available from https://imagej.net/MyofibrilJ. Raw data used to generate all plots presented in figure panels are available in the source data files for *Figures 1*, *5*, *6*, *7* and *8*. Data on statistical test results are presented in *Supplementary file 5*.

## Acknowledgements

We thank Irene Ferreira for constructing the tagged *kon-tiki* allele and the Bloomington and VDRC stock centers for fly stocks. We are grateful to Reinhard Fässler and Andreas Ladurner for generous support and to Bettina Stender for excellent technical assistance. We thank Elena Nikonova for initial analysis of the *Act88F*-GAL4 x *salmIR* phenotype and Sandra Esser for testing the *salm* KK181052 hairpin. We acknowledge the VBCF (Vienna, AT) for mRNA-sequencing and the Core Facility Bioimaging at the Max Planck Institute for Biochemistry and the LMU Biomedical Center (Martinsried, DE) for help with confocal image analysis. We thank Florian Marty for help with the BioSorter, and Alexander Stark for help setting-up the mRNA-sequencing. We thank Aynur Kaya-Copur and Wouter Koolhaas for helpful discussions, and Nuno Luis and Vincent Loreau for insightful comments on the manuscript. Our work was supported by the Max Planck Society and the CNRS; postdoctoral Humboldt, EMBO long-term (688–2011), and NIH-NRSA (5F32AR062477) fellowships (MLS), the Frederich-Bauer Stiftung (MLS), the Center for Integrated Protein Science München (MLS), the Boehringer Ingelheim Fonds (SBL), a Career Development Award from the Human Frontier Science Program (FS), the EMBO Young Investigator Program (FS) and the European Research Council under the European Union's Seventh Framework Programme (FP/2007–2013)/ERC Grant 310939 (FS), the excellence initiative Aix-Marseille University AMIDEX (ANR-11-IDEX-0001–02) (FS), the ANR-ACHN (FS) and the LabEX-INFORM (ANR-11- LABX-0054) (FS) and France Bioimaging (ANR-10-INBS-04–01) (FS).

## Additional information

### Funding

| Funder | Grant reference number | Author |
|---|---|---|
| Max-Planck-Gesellschaft | | Maria L Spletter<br>Christiane Barz<br>Assa Yeroslaviz<br>Xu Zhang<br>Sandra B Lemke<br>Giovanni Cardone<br>Bianca H Habermann<br>Frank Schnorrer |
| European Molecular Biology Organization | EMBO-LTR 688-2011 | Maria L Spletter |
| Alexander von Humboldt-Stiftung | | Maria L Spletter |
| National Institute for Health Research | 5F32AR062477 | Maria L Spletter |
| Centre National de la Recherche Scientifique | | Xu Zhang<br>Adrien Bonnard<br>Bianca H Habermann<br>Frank Schnorrer |
| Boehringer Ingelheim Fonds | | Sandra B Lemke |
| Center for Integrated Protein Science München | the Boehringer Ingelheim Fonds | Sandra B Lemke |
| Aix-Marseille Université | ANR-11-IDEX-0001-02 | Bianca H Habermann<br>Frank Schnorrer |
| H2020 European Research Council | ERC Grant 310939 | Frank Schnorrer |
| Agence Nationale de la Recherche | ANR-11- LABX-0054 | Frank Schnorrer |
| European Molecular Biology Organization | EMBO-YIP | Frank Schnorrer |
| Agence Nationale de la Recherche | ANR-10-INBS-04- 01 | Frank Schnorrer |

| Agence Nationale de la Re-cherche | ANR ACHN | Frank Schnorrer |

The funders had no role in study design, data collection and interpretation, or the decision to submit the work for publication.

## Author contributions

Maria L Spletter, Data curation, Formal analysis, Validation, Investigation, Visualization, Methodology, Writing—original draft, Writing—review and editing; Christiane Barz, Data curation, Investigation; Assa Yeroslaviz, Software, Formal analysis, Visualization, Methodology, Writing—review and editing; Xu Zhang, Sandra B Lemke, Data curation, Formal analysis, Investigation, Writing—review and editing; Adrien Bonnard, Formal analysis, Investigation, Visualization; Erich Brunner, Investigation, Methodology; Giovanni Cardone, Software, Writing—review and editing; Konrad Basler, Funding acquisition, Methodology, Writing—review and editing; Bianca H Habermann, Software, Formal analysis, Funding acquisition, Validation, Visualization, Methodology, Writing—review and editing; Frank Schnorrer, Conceptualization, Supervision, Funding acquisition, Validation, Writing—original draft, Writing—review and editing

## Author ORCIDs

Maria L Spletter [iD] http://orcid.org/0000-0002-2068-3350
Xu Zhang [iD] http://orcid.org/0000-0002-1628-9895
Sandra B Lemke [iD] http://orcid.org/0000-0002-3817-4351
Giovanni Cardone [iD] http://orcid.org/0000-0003-4712-1451
Bianca H Habermann [iD] http://orcid.org/0000-0002-2457-7504
Frank Schnorrer [iD] http://orcid.org/0000-0002-9518-7263

## Decision letter and Author response

Decision letter https://doi.org/10.7554/eLife.34058.051
Author response https://doi.org/10.7554/eLife.34058.052

## Additional files

### Supplementary files

• Supplementary file 1. mRNA-Seq raw data The file includes multiple tabs containing the raw or input counts data from bioinformatics analysis, as well as a key to all original data provided in the supplementary tables. This table includes mRNA-Seq counts data, DESeq2 normalized counts data and standard normal counts data used for Mfuzz clustering for wild-type and *salmIR* IFM time points. The averaged core expression profiles for each Mfuzz cluster are also listed.
DOI: https://doi.org/10.7554/eLife.34058.040

• Supplementary file 2. GO-Elite analysis data. This table includes multiple tabs containing the GO-Elite analysis of enrichments in Mfuzz clusters as well as genes up- or down-regulated from 30–72 hr APF and between wild-type and *salmIR* IFM. It also contains a complete list of all genes included in the 'User Defined' gene sets.
DOI: https://doi.org/10.7554/eLife.34058.041

• Supplementary file 3. Summary of sarcomere and myofibril quantifications This table includes a numerical summary of quantification values reported graphically in *Figures 5*, *7* and *8*. Quantifications of sarcomere length, myofibril width and myofibril diameter were performed with the MyofibrilJ script (see Materials and methods). Fiber length and cross-sectional area measurements were performed in Fiji/Image J.
DOI: https://doi.org/10.7554/eLife.34058.042

• Supplementary file 4. DESeq2 pairwise differential expression analysis This table contains multiple tabs containing the output data from DESeq2 differential expression analysis between sequential IFM development time points, from 30 to 72 hr APF as well as between WT and *salmIR* IFM.
DOI: https://doi.org/10.7554/eLife.34058.043

• Supplementary file 5. Statistical Data This table includes multiple tabs containing the statistical and calculation data for the different panels of *Figures 1*, *5*, *6*, *7* and *8*.
DOI: https://doi.org/10.7554/eLife.34058.044

• Transparent reporting form
DOI: https://doi.org/10.7554/eLife.34058.045

## Data availability

Processed data from DESeq2, Mfuzz and GO-Elite are available in Supplementary Files 1, 2, 4. mRNA-Seq data are publicly available from NCBI's Gene Expression Omnibus (GEO) under accession number GSE107247. Fiji scripts for analysis of sarcomere length, myofibril width and myofibril diameter are available from https://imagej.net/MyofibrilJ. Raw data used to generate all plots presented in figure panels are available in the source data files for Figures 1, 5, 6, 7 and 8. Data on statistical test results are presented in Supplementary File 5.

The following dataset was generated:

| Author(s) | Year | Dataset title | Dataset URL | Database, license, and accessibility information |
|---|---|---|---|---|
| Spletter ML, Schnorrer F, Yeroslaviz A, Stark A, Habermann B | 2017 | Systematic transcriptomics reveals a biphasic mode of sarcomere morphogenesis in flight muscles regulated by Spalt | https://www.ncbi.nlm.nih.gov/geo/query/acc.cgi?acc=GSE107247 | Publicly available at the NCBI Gene Expression Omnibus (accession no: GSE10 7247). |

The following previously published dataset was used:

| Author(s) | Year | Dataset title | Dataset URL | Database, license, and accessibility information |
|---|---|---|---|---|
| Spletter ML, Schnorrer F, Gerlach D, Stark A, Yeroslaviz A, Habermann B | 2014 | The RNA binding protein Arrest (Aret) regulates myofibril maturation in Drosophila flight muscle | https://www.ncbi.nlm.nih.gov/geo/query/acc.cgi?acc=GSE63707 | Publicly available at the NCBI Gene Expression Omnibus (accession no: GSE63707). |

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
