## [Decision Letter]

Thank you for submitting your article "Systematic transcriptomics reveals a biphasic mode of sarcomere morphogenesis in flight muscles regulated by Spalt" for consideration by *eLife*. Your article has been reviewed by three peer reviewers, and the evaluation has been overseen by K VijayRaghavan as the Senior/Reviewing Editor. The reviewers have opted to remain anonymous.

The reviewers have discussed the reviews with one another and the Reviewing Editor has drafted this decision to help you prepare a revised submission.

Summary:

Systematic transcriptomics reveals a biphasic mode of sarcomere morphogenesis in flight muscles regulated by Spalt by Spletter et al., addresses an important issue of global gene expression underlying development of *Drosophila* indirect flight muscle (IFM), which represent an attractive model for multi-fibrillar muscle formation in general. Even if developmental time points used for transcriptomics cover the entire myogenic process, authors focus here mainly on myofibrillogenesis and sarcomerogenesis that take place from 30 h APF until the end of pupa stage. Using dedicated bioinformatics tools they identify a large number of gene clusters that display similar temporal profiles of expression and among them clusters No. 13 and No. 22 enriched in genes encoding sarcomeric components. For example genes from cluster 22, characterized by an increase in expression between 48 and 72h, include Mhc, Act88F and Mf. In parallel, authors present a detailed view of different steps of IFM formation largely based on in vivo imaging of IFM development and on expression analyses of sarcomeric components at protein levels. Based on transcriptional profiling showing that expression of a number of sarcomeric transcripts peak at 72 h APF but also on the observations that some sarcomeric components (Mhc, Obscurin) could be detected at protein levels as early as at 30 h APF authors propose that sarcomerogenesis involves 2 phases: i) formation of immature sarcomeres; ii) sarcomere maturation.

Authors also try to correlate their time-course transcriptional data in the developing IFMs with gene deregulation in the context of late Salm knockdown and conclude that Salm in addition to its role in the specification of fibrillar muscle fate is also required for the sarcomere maturation in flight muscles. Altogether, the manuscript is rich in large-scale data and well illustrated providing many quantitative assessments of developing myofibrils and sarcomeres over time. It could thus represent an important resource for the community of muscle research. This paper is also technically sound (but see specific major comments) and will be an excellent resource for scientists interested in muscle development. Unfortunately, it wants to be more than a resource paper, and the statements in the title are not supported by evidence. An important concern is the author's interpretation of sarcomere formation that they propose to be biphasic and several of our points relate to this is different ways. Further, one gets the feeling that the *salm* data was simply added, because this was already a familiar gene to the authors. The *stretchin* data (Figure 7 and 8) are also somewhat unmotivated: *stretchin* knockdown has a stronger phenotype than *salm* knockdown, even though *salm* is supposed to upregulate dozens of sarcomeric genes in addition to *stretchin*.

In summary, these and the points below lead us to the conclusion that as a resource, this paper is great, and there are many hidden gems, e.g. the down-regulation of microtubule genes after the sarcomere formation phase (after 30h APF), which could indicate an involvement of MTs in setting up initial myofibril polarity as suggested by Reedy and Beall. However, the authors try to over- interpret their results and the title and interpretation are somewhat disconnected with what the data allows.

Given the value as a resource, the authors should substantially re-write the paper accordingly. It does not make it any less of value, indeed makes it value greater.

Essential revisions:

The first three points address the same issue.

1) When looking at all the presented data one could rather conclude that sarcomere formation is a continuous process tightly correlated with myofibrillogenesis and involving 3 distinct steps: a. simultaneous formation of first set of immature sarcomeres (here 80) during compaction force-driven formation of myofibrils at 30 h APF. b. Increasing myofibrils length by adding of immature sarcomeres (up to 230) between 30 and 48 h APF – here according to the scheme (Figure 9) immature sarcomeres are already more complex and longer than those formed initially; c. Increasing myofibrils length and diameter by incorporating to the sarcomeres of additional “core” components (e.g. Mhc) and muscle-type specific components (e.g. Mf, Fln) Thus, the first phase authors call “sarcomere formation” between (30 and 48 h APF), already includes some maturation events. Also, the model presented in Figure 9 fits rather into a 3 step continuous process.

2) In a slightly different perspective on the 'biphasic expression': As mentioned above, both the transcriptional data and bioinformatics analysis are we'll done. However, despite the accurate bioinformatics and clustering data there is not enough novelty regarding the difference between the two phases of flight muscle development and how they are regulated. The data presented are based on correlations between the distinct clusters and distinct stages of muscle maturation and remain at the level of parallel observations. As the authors discuss, many of the observations have already been previously described, even by the authors themselves. The link between myofibrillogenesis and transcription of muscle proteins is trivial. Does transcription at the second phase depends on muscle mechanics, or it only represents the need for the muscle to grow in size. Are distinct factors control the first versus the second phase? Moreover, as the authors note in their Discussion, large scale transcriptomic changes have already reported during mouse, pig, and fish skeletal muscles growth, so the additional novelty of the present study is not entirely clear. The observation that Salm is required for the late stage of sarcomere maturation and growth is interesting, however, the authors did not elaborate on how the same transcription factor promotes two distinct phases of sarcomere development. Is it the levels of Salm, or its putative cooperation with additional factors that are present only in the second phase? Could it be that the differential mechanical inputs together with Salm transcriptional input change the outcome?

3) And in a third view on the same theme: The authors distinguish a sarcomere formation phase followed by a sarcomere maturation phase. They give as an example cluster 22, which they say is strongly up-regulated in the maturation phase. What we can see is a gradual up-regulation of all of these genes starting at around 24h-30h APF (sarcomere formation) with a peak shortly before hatching, which corresponds precisely to the progression of muscle development as laid out in Figure 1, and is not surprising. To make all this muscle mass, massive transcription of cytoskeletal genes is required. To spell out one example, the Mhc gene: according to Supplementary file 1, sheet 2, Mhc at 30h (570173) is already 600-fold up-regulated compared to 16h (931). That is, Mhc is very strongly expressed when sarcomeres form (as it should be), and becomes even more massively expressed, when myofibrils grow in length and volume. The same applies to most other sarcomeric genes. The authors say this biphasic mode is regulated by *salm*. Or, to rephrase, the massive up-regulation from 48-90h APF is due to *salm*. This does not fit with their own data. Salm has an expression peak at 72h APF (cluster 26), too late to cause the massive up-regulation starting at 30h APF, and correspondingly, their knock-down data (Figure 6) are unconvincing. The authors stress that this is likely a general mechanism of muscle development, but provide no evidence. Rather, one of the interesting effects that *salm* knockdown has (muscle contractions do not stop at 72h in *salm* knockdown IFM in Figure 7) is clearly specific to IFM, because other muscles (embryonic or abdominal) contract increasingly strongly throughout development.

4) Figure 1 – No legend to schemes presenting different steps of DLM development (last column) is provided. What does the blue drawing represent? If this corresponds to Shot expression, its presence on the scheme at 16 h APF does not fit with the immunostaining (B').

5) Figure 1 – How was the length of wavy fibers at 90 h APF determined;

6) Figure 2 and Figure 2—figure supplement 2 – temporal Aret expression profile (cluster 14) appears different from that of its activator Salm (cluster 26).

7) Figure 2—figure supplement 2 – Salm transcripts level are up-regulated at 30h but down at 48h. At protein levels 30h is not presented but immunostaining at 48 h shows high protein accumulation – higher than at 72 h when the mRNA peak is highest. The authors need to comment this discrepancy – could it be related to technical issues/quality of biological sample for transcriptional profiling?

8) Figure 2. – Also, RNA profile and protein detection of Talin do not correlate. βPS-integrin seems to be a better example and could move to the main figure instead of Talin.

9) Figure 2—figure supplement 1 – Number of different temporal profiles – 40 clusters appears high. This is certainly related to constrain when using Mfuzz. Some of clusters look similar. Did authors try to analyse their data with less clusters?

10) Figure 3 is not particularly important for the data analysis and could be moved to supplementary data.

11) Figure 5—figure supplement 1 – How do the authors explain detection of Mhc and Obscurin at 30h in the sarcomeres even if their transcripts are not detected (or have we missed something here?)?

12) Figure 7 and 8 – Strn-Mlck appears to be the main target of Salm during the maturation phase, Strn-Mlck expression is reduced in sarcomeres from Act88F>salmIR pupae, however effects of Salm loss of function leads to a reduced sarcomere size while in Strn-Mlck mutants at 90 h APF sarcomeres are much longer. These differences of phenotypes are surprising and suggest that Strn-Mlck is not the major Salm target in the second phase. Did authors test other potential candidates whose function fits better with that of late Salm phenotypes?

13) Figure 9 – Use in the scheme only sarcomeric components analysed in this work or refer to published data if you include others.

14).Other comments: In Supplementary file 1, sheet 1, the cluster assignment is incorrect: e.g. unc-89 is listed as cluster 24 (should be 13), Mhc is listed as cluster 39 (should be 22), etc.

15) Figure 1 and 5 are beautiful and quantitatively very rigorous, but essentially repeat what we know since Reedy and Beall (1993) and more recent publications from the Schnorrer lab.

---

## [Author Response]

[…] Unfortunately, it wants to be more than a resource paper, and the statements in the title are not supported by evidence. An important concern is the author's interpretation of sarcomere formation that they propose to be biphasic and several of our points relate to this is different ways.

We agree with the reviewers’ comment that the biphasic model was an over-simplification of our observations and we apologize for that. A more accurate interpretation of our observations is 1) around 80 immature sarcomeres assemble simultaneously in each myofibril at about 30 h APF; 2) around 200 new sarcomeres are added to each immature myofibril as they grow and develop until about 48 h APF; 3) subsequently, as the myofibrils/sarcomeres continue to mature, they grow significantly in length and diameter, but *no* new sarcomeres are being added (at least not after 60 h APF).

Taking this into account, we now define three ordered but also continuous phases, so we now call them sequential but overlapping phases 1) simultaneous sarcomere assembly phase at 30-34 h APF; 2) sarcomere addition/maturation phase until shortly after 48 h and 3) sarcomere growth and maturation phase until adulthood.

Importantly, our work discovered that new myofibrils are only assembled at 30 h APF: we quantified the number of myofibrils per muscle fiber, which remains constant after 30 h APF. This suggests that the adult fly is equipped with myofibrils which all originate from the first simultaneous assembly phase. Hence, we believe it is significant to define these sequential phases, making our manuscript different from a classical 'resource paper'. However, as the reviewers and editor suggested, we also find our paper appropriate for the *eLife* 'resource section'. We have adjusted the title and content accordingly and hope the new title faithfully represents the results.

Further, one gets the feeling that the salm data was simply added, because this was already a familiar gene to the authors.

Our work identified a transcriptional transition after 30 h APF, with many genes including two clusters (13 + 22) enriched for sarcomere protein coding genes, whose expression is boosted after 30 h APF. To understand these transcriptional dynamics mechanistically, we tested a number of candidate regulators, including Mef2, E2F, Salm and Him, however only Salm led to conclusive results. We show that Salm is required for full induction of the transcriptional transition after 30 h APF at the systems level. We have verified the Salm-dependent induction of a number of sarcomeric proteins, including Stretchin, Flightin and Mhc, at the protein level. As this was performed using quantitative microscopy of protein fusions under endogenous control, we believe our data significantly supports that the observed transcriptional transition is at least in part regulated by Salm. Thus, we feel the *salm* RNAi data do fit well to the rest of the manuscript and provide a first mechanistic explanation of parts of the transcriptional dynamics observed. However, we agree with the reviewers that Salm is not the only regulator as the transition is not entirely blocked in *salm* RNAi, and thus we adjusted the title and statements throughout the manuscript to make clear that Salm as well as other genes contribute to the regulation. This is also addressed more clearly in the Discussion.

We have further updated our curated ‘sarcomere protein list’ to more accurately represent structural sarcomeric proteins, as the previous list contained some non-sarcomeric genes, such as microtubule binding proteins and RNA binding proteins, that potentially influence sarcomere assembly. The new list is included in Supplementary file 2, and all of the bioinformatics panels and enrichment analyses throughout the manuscript have been updated to reflect the new list.

The stretchin data (Figure 7 and 8) are also somewhat unmotivated: stretchin knockdown has a stronger phenotype than salm knockdown, even though salm is supposed to up-regulate dozens of sarcomeric genes in addition to stretchin.

Stretchin is one of the strongest induced mRNAs after 30 h APF. It is first detectable on the protein level at 48 h APF and is extremely high expressed at 72 h APF onwards. Thus, it is not only connected to the observed transcriptional transition, but a bone fide example of it. As the reviewers point out correctly, Salm regulates many sarcomeric genes. This is likely the explanation why the *stretchin* loss of function phenotype is different to the *salm* RNAi phenotype. However, the *stretchin* loss of function phenotype is *not* stronger than the *salm* RNAi phenotype. In *salm* RNAi, sarcomere growth is reduced already after 48 h APF, whereas in the *stretchin* mutant the sarcomeres overgrow at 80 h APF and only later muscle hypercontraction results in too short sarcomeres in the adult *stretchin* mutants. We made this difference more clear in the revised version of the manuscript. However, we believe that both aspects, the *salm* RNAi and the *stretchin* mutant data, fit very well to the transcriptional transition of the sarcomeric protein coding genes identified in the first part of the manuscript.

In summary, these and the points below lead us to the conclusion that as a resource, this paper is great, and there are many hidden gems, e.g. the down-regulation of microtubule genes after the sarcomere formation phase (after 30h APF), which could indicate an involvement of MTs in setting up initial myofibril polarity as suggested by Reedy and Beall. However, the authors try to over- interpret their results and the title and interpretation are somewhat disconnected with what the data allows.Given the value as a resource, the authors should substantially re-write the paper accordingly. It does not make it any less of value, indeed makes it value greater.

We are happy that the reviewers appreciate the importance of our study as a resource. Apart from the early microtubule cluster or various mitochondrial clusters, our data for example also contains expression dynamics of all RNA binding protein or transcription factors. We have decided to focus on the two clusters enriched for sarcomere protein coding genes in this manuscript, and thus included the biological findings related to these, which we would like to keep together. As explained above, we have reworked our manuscript including its interpretation and hope that this revised version appears suitable for publication in *eLife* as a resource with strong biological implications.

Essential revisions:The first three points address the same issue.1) When looking at all the presented data one could rather conclude that sarcomere formation is a continuous process tightly correlated with myofibrillogenesis and involving 3 distinct steps: a. simultaneous formation of first set of immature sarcomeres (here 80) during compaction force-driven formation of myofibrils at 30 h APF. b. Increasing myofibrils length by adding of immature sarcomeres (up to 230) between 30 and 48 h APF – here according to the scheme (Figure 9) immature sarcomeres are already more complex and longer than those formed initially; c. Increasing myofibrils length and diameter by incorporating to the sarcomeres of additional “core” components (e.g. Mhc) and muscle-type specific components (e.g. Mf, Fln) Thus, the first phase authors call “sarcomere formation” between (30 and 48 h APF), already includes some maturation events. Also, the model presented in Figure 9 fits rather into a 3 step continuous process.

We thank the reviewers for this point to which we entirely agree. As explained above, we have reworked our manuscript to now present 3 sequential but overlapping phases. By definition, 3 phases need to be somewhat distinct from each other – otherwise it would only be one phase. However, the 3 phases are connected and there is no sharp switch, thus we call them now sequential phases. We incorporated these points into the revised version of our manuscript.

2) In a slightly different perspective on the 'biphasic expression': As mentioned above, both the transcriptional data and bioinformatics analysis are we'll done. However, despite the accurate bioinformatics and clustering data there is not enough novelty regarding the difference between the two phases of flight muscle development and how they are regulated. The data presented are based on correlations between the distinct clusters and distinct stages of muscle maturation and remain at the level of parallel observations. As the authors discuss, many of the observations have already been previously described, even by the authors themselves.

We believe that not only the systematic transcriptomics but also the sequential phases of myofibrillogenesis are first described in quantitative detail in this manuscript. We quantified the length of the muscle fibers, the sarcomeres and consequently the number of sarcomeres per myofibril much more carefully than earlier work by Reedy and Beall (1993), which was done at 22 degrees, so is also difficult to relate to our transcriptomics time‐points. In particular, the cross sections in which we have also analyzed myofiber (muscle) diameter fueled the discovery that myofibrils are only built once, at 30h APF. In turn, we conclude that all myofibrils in the adult must progress through the sequential phases during development. Thus, our manuscript contains novel data that sparked novel conclusions, which in our opinion are important points of the manuscript.

The link between myofibrillogenesis and transcription of muscle proteins is trivial. Does transcription at the second phase depends on muscle mechanics, or it only represents the need for the muscle to grow in size. Are distinct factors control the first versus the second phase? Moreover, as the authors note in their Discussion, large scale transcriptomic changes have already reported during mouse, pig, and fish skeletal muscles growth, so the additional novelty of the present study is not entirely clear. The observation that Salm is required for the late stage of sarcomere maturation and growth is interesting, however, the authors did not elaborate on how the same transcription factor promotes two distinct phases of sarcomere development. Is it the levels of Salm, or its putative cooperation with additional factors that are present only in the second phase? Could it be that the differential mechanical inputs together with Salm transcriptional input change the outcome?

We have not tested if the transcriptional change after 30 h APF depends on muscle forces/mechanics. This is certainly an interesting aspect to investigate in the future. We thank the reviewers for pointing it out, but we believe it is beyond the scope of our current manuscript.

Transcriptional dynamics of some other muscles systems have been reported (much less quantitatively and systematically, as compared to our report here, and mainly restricted to postembryonic muscle growth), but very little is known about the mechanism instructing the change in transcription. We attempted to address this by investigating the role of Salm during the transcriptional transition. In the future, it will be exciting to see if this part of Salm’s function requires contractile muscle force or not, as well as to identify additional factors regulating other stages of myofibril development. We have added this idea to the Discussion part of our manuscript, where we already discuss the possibility that Salm may cooperate with additional factors, such as E2F, which is also required at this late stage of muscle development (Zappia and Frolov, 2016).

3) And in a third view on the same theme: The authors distinguish a sarcomere formation phase followed by a sarcomere maturation phase. They give as an example cluster 22, which they say is strongly up-regulated in the maturation phase. What we can see is a gradual up-regulation of all of these genes starting at around 24h-30h APF (sarcomere formation) with a peak shortly before hatching, which corresponds precisely to the progression of muscle development as laid out in Figure 1, and is not surprising. To make all this muscle mass, massive transcription of cytoskeletal genes is required. To spell out one example, the Mhc gene: according to Supplementary file 1, sheet 2, Mhc at 30h (570173) is already 600-fold up-regulated compared to 16h (931). That is, Mhc is very strongly expressed when sarcomeres form (as it should be), and becomes even more massively expressed, when myofibrils grow in length and volume. The same applies to most other sarcomeric genes.

The reviewer is correctly pointing out that Mhc is already expressed at 30 h APF (in contrast to 16 h APF). As said, this was expected, as myofibrils assemble at 30 h APF. However, it was much less expected, at least for us, to find the strong induction after 30 h APF, in particular as all myofibrils are already present at 30 h APF. There is a 2-fold log_2_FC for Mhc from 24h to 30h, but another 3-fold log_2_FC from 48h to 72h (so as strong if not stronger induction at the later stage). Additionally, some sarcomeric protein coding genes, like Stretchin, Myofilin and Flightin, are only expressed after 30 h APF and the proteins are not detectable at 30 h APF, at least not with our GFP fusion proteins expressed under endogenous control. See also the new data in Figure 6—figure supplement 3 quantifying the level of protein expression at the same developmental stages. Thus, the transition after 30 h APF is an active process (and not just doubling or tripling of mRNAs that are present before), which at least for us is interesting, novel and unexpected. Additionally, we linked the transcriptional transition to the functional change of gaining stretch activatability in flight muscles. Thus, the transition clearly has functional consequences.

The authors say this biphasic mode is regulated by salm. Or, to rephrase, the massive up-regulation from 48-90h APF is due to salm. This does not fit with their own data. Salm has an expression peak at 72h APF (cluster 26), too late to cause the massive up-regulation starting at 30h APF, and correspondingly, their knock-down data (Figure 6) are unconvincing.

The transcriptional dynamics of Salm are consistent with a role of Salm in the observed transcriptional transition after 30 h APF. We showed with semi-quantitative imaging of Salm protein that Salm protein is expressed at 17 h APF, and protein levels strongly increase until 48 h APF, before they get down-regulated at 72 h APF (Figure 2—figure supplement 2C‐C’’). These protein dynamics are also consistent with a *salm* mRNA peak at 30 h APF (protein is high at 48 h APF). The mRNA peak at 72 h APF may indicate a role even later in the process, which we have not investigated (see also comment below). Obviously, protein expression can be regulated post-translationally, hence confirmation at the protein level is important. In summary, we show high levels of *salm* mRNA at 30 h and of Salm protein at 48 h, both of which are consistent with a role of Salm in the transcriptional transition after 30 h APF. This transition is attenuated in *salm* RNAi muscle. We showed this bioinformatically using mRNA data at the systems level, and individually for selected candidates at the protein level (Figure 6). To make the bioinformatics in Figure 6 visually clearer, we have added violin plots to more clearly illustrate the decrease in expression of particular gene sets in *salmIR* and to emphasize that many of these genes, from cluster 22 for example, also fail to be induced as strongly from 30 h to 72 h APF compared to wild type. Further, to make our fluorescent protein expression data more convincing, we have added Figure 6—figure supplement 3, in which we detail time-course expression data for Mhc-GFP, Unc-89-GFP, Fln-GFP and Strn-Mlck-GFP using the fluorescent bead assay. These data quantify the strong increase in protein expression from 30 h to 72 h APF for these proteins, as well as provide a basis to judge how significant the decrease in protein expression is in *salmIR* muscle. We also show that a decrease in expression in the case of *Mef2*-GAL4>Gma-GFP is detectable. We believe these additions should make the knock-down data in Figure 6 more convincing. However, we do agree that Salm is not inducing the transition in expression alone, as there is still transcriptional up-regulation in the absence of Salm. We have adjusted the text and the title accordingly in the revised manuscript.

The authors stress that this is likely a general mechanism of muscle development, but provide no evidence. Rather, one of the interesting effects that salm knockdown has (muscle contractions do not stop at 72h in salm knockdown IFM in Figure 7) is clearly specific to IFM, because other muscles (embryonic or abdominal) contract increasingly strongly throughout development.

We suggest that the observed transcriptional transition during sarcomere development corresponding to sarcomere maturation is a general mechanism, as for example during vertebrate muscle development there is a reported transcriptional switch from embryonic to fetal myosin genes at specific stages, which likely influence sarcomere function and structure (Schiaffino and Reggiani, 2011). We are now more specific on this point in the Discussion of our revised manuscript.

The specific *salm*‐dependent gain of stretch-activation, which we have shown here for the first time, is indeed a flight muscle-specific feature. We make no claim that this particular change is general. However, the vertebrate heart also does acquire significant stretch activation capacity during development, as described by the Frank-Starling mechanism of heart contraction (Shiels et al. J. Exp Biol. 2008).

4) Figure 1 – No legend to schemes presenting different steps of DLM development (last column) is provided. What does the blue drawing represent? If this corresponds to Shot expression, its presence on the scheme at 16 h APF does not fit with the immunostaining (B').

We apologize for missing description of the schemes in the legend. In blue are the tendon cells, which do express Shot protein, but as the reviewer pointed out, Shot is also expressed in the muscle. We have modified the legend accordingly.

5) Figure 1 – How was the length of wavy fibers at 90 h APF determined;

As described in the Materials and methods section, we have traced the edges of the myofiber from one attachment site to the other using the freehand tool in Fiji.

6) Figure 2 and Figure 2—figure supplement 2 – temporal Aret expression profile (cluster 14) appears different from that of its activator Salm (cluster 26).

Both *aret* and *salm* show an mRNA expression peak at 30 h APF. We show that Salm protein is already expressed at 17 h APF, at which stage only little of Aret protein is found compared to 24 h APF (Figure 2D-F, Figure 2—figure supplement 2A-C). This is consistent with Salm protein regulating *aret* mRNA expression (as shown in Spletter et al., 2015).

7) Figure 2—figure supplement 2 – Salm transcripts level are up-regulated at 30h but down at 48h. At protein levels 30h is not presented but immunostaining at 48 h shows high protein accumulation – higher than at 72 h when the mRNA peak is highest. The authors need to comment this discrepancy – could it be related to technical issues/quality of biological sample for transcriptional profiling?

We have no information about Salm protein turn-over rates at this point. It is conceivable that Salm protein stability may change after 48 h APF, explaining the drop in protein levels. Generally, in our mRNA-seq data transcription factors are expressed at much lower read-counts than structural proteins, hence these data can be indeed more variable. We have added a comment to the manuscript suggesting that the peak at 72 h of *salm* mRNA would need to be verified.

8) Figure 2. – Also, RNA profile and protein detection of Talin do not correlate. βPS Integrin seems to be a better example and could move to the main figure instead of Talin.

We agree with the reviewer and have switched Talin with βPS-Integrin to illustrate the point better in the main part of the figure. Thanks.

9) Figure 2—figure supplement 1 – Number of different temporal profiles – 40 clusters appears high. This is certainly related to constrain when using Mfuzz. Some of clusters look similar. Did authors try to analyse their data with less clusters?

We comment on this issue in the Materials and methods section of the manuscript. We have systematically tested Mfuzz clustering with cluster numbers ranging from 5 to 250. After extensive tests, we decided that 40 is the best compromise to represent the complexity of the transcriptional dynamics and distinct clusters are well resolved. A lower number of clusters results in a merging of clusters, whose individual gene members show quite distinct temporal profiles.

10) Figure 3 is not particularly important for the data analysis and could be moved to supplementary data.

We agree with the reviewer that Figure 3 is not essential for the data analysis. However, if space allows we would like to keep it in the main text, as it provides a good overview for the general reader, who is interested in other aspects of muscle biology apart from sarcomeres. In particular, for the ‘resource section’ of *eLife*, this figure would be important, as it amongst other things illustrates the dynamics of the early’ microtubule clusters’ or the various mitochondrial pathways for the general reader.

11) Figure 5—figure supplement 1 – How do the authors explain detection of Mhc and Obscurin at 30h in the sarcomeres even if their transcripts are not detected (or have we missed something here?)?

As discussed above, Mhc and Obscurin mRNAs are expressed at 30 h APF, but compared to 48 h APF the levels are low. This is illustrated by the small increase in the slope of the mRNA profile plots in Figure 2Q and Figure 2—figure supplement 2K from 16 h to 24 h to 30 h APF. It is also clear from Supplementary file 1 that contains all the mRNA read counts, that Mhc and Obscurin mRNA are present at a low level already at 30 h APF. We have adjusted the figure legend to make this more clear in the revised version.

This confusion illustrates well why we are so excited about this strong transcriptional induction, which we did not expect. Expression of many sarcomeric protein coding genes at 48 h APF is so high that the existing expression at 30 h APF is hard to see even on the log-scale plot profile that we show.

12) Figure 7 and 8 – Strn-Mlck appears to be the main target of Salm during the maturation phase, Strn-Mlck expression is reduced in sarcomeres from Act88F>salmIR pupae, however effects of Salm loss of function leads to a reduced sarcomere size while in Strn-Mlck mutants at 90 h APF sarcomeres are much longer. These differences of phenotypes are surprising and suggest that Strn-Mlck is not the major Salm target in the second phase. Did authors test other potential candidates whose function fits better with that of late Salm phenotypes?

Stretchin is indeed one of many genes that are regulated by Salm after 30 h, thus removing a single target should not easily recapitulate the phenotype of the regulator. As sarcomere length is always a balance between thin and thick filament growth, as well as thin and thick filament overlap, it is not simple to relate sarcomere length differences directly to molecular differences. After eclosion, muscle fibers are hyper-contracted in both genotypes (*salm* RNAi and *stretchin* mutants), and sarcomeres are indeed too short in both genotypes at this stage.

We have not extensively tested other sarcomeric proteins for their function after 30 h APF.

13) Figure 9 – Use in the scheme only sarcomeric components analysed in this work or refer to published data if you include others.

We thank the reviewer for pointing this out. We added additional references.

14) Other comments: In Supplementary file 1, sheet 1, the cluster assignment is incorrect: e.g. unc-89 is listed as cluster 24 (should be 13), Mhc is listed as cluster 39 (should be 22), etc.

We apologize for this mistake. The table was from an earlier version. We fixed it in the revised version and thank the reviewer for finding our mistake.

15) Figure 1 and 5 are beautiful and quantitatively very rigorous, but essentially repeat what we know since Reedy and Beall (1993) and more recent publications from the Schnorrer lab.

As discussed above, these careful quantifications, including the cross-sections at the corresponding stages, were essential for a key discovery of the manuscript – myofibrils are assembled only once and their number remains constant after 30 h APF. This finding sparked the idea of the subsequent sarcomerogenesis/ myofibrillogenesis phases, since all myofibrils are made initially, then need to grow by adding more sarcomeres, and finally individual sarcomeres grow in length and diameter. Such an analysis had not been done before, however we certainly do cite the earlier work that inspired our new studies.

Additionally, we present a novel script for semi-automated analysis of sarcomere length and myofibril width, allowing us to trace changes in these parameters at a higher resolution and across more time points than previously reported. Lastly, we present a completely novel finding in Figure 5I-J that IFMs spontaneously contract during development but lose this ability by 72 h APF. This finding more broadly impacts how we think about myofibril development and the establishment of stretch-activation in muscle fibers.